

# Potential impacts of marine fuel regulations on Arctic clouds and radiative feedbacks

Luís Filipe Escusa dos Santos[1,a], Hannah C. Frostenberg[2], Alejandro Baró Pérez[2], Annica M. L. Ekman[3,4], Luisa Ickes[2], and Erik S. Thomson[1]

[1]Department of Chemistry and Molecular Biology, University of Gothenburg, Gothenburg, Sweden.
[2]Department of Space, Earth and Environment, Chalmers University of Technology, Gothenburg, Sweden.
[3]Department of Meteorology, Stockholm University, Stockholm, Sweden.
[4]Bolin Centre for Climate Research, Stockholm, Sweden.
[a]now at: Department of Space, Earth and Environment, Chalmers University of Technology, Gothenburg, Sweden.

**Correspondence:** Luis F. E. d. Santos (luis.santos@chalmers.se) and Luisa Ickes (luisa.ickes@chalmers.se)

**Abstract.** Increased surface warming over the Arctic, triggered by increased greenhouse gas concentrations and feedback processes in the climate system, has been causing a steady decline in sea-ice extent and thickness. With the retreating sea-ice, shipping activity will likely increase in the future driven by economic activity and the potential for realizing time and fuel savings from transiting shorter trade routes. Moreover, over the last decade, the global shipping sector has been subject to reg-

ulatory changes, that affect the physicochemical properties of exhaust particles. International regulations aiming to reduce $SO_x$ and particulate matter (PM) emissions, mandate ships to burn fuels with reduced sulfur content or alternatively, use wet scrubbing as exhaust after-treatment when using fuels with sulfur contents exceeding regulatory limits. Compliance measures affect the physicochemical properties of exhaust particles and their cloud condensation nuclei (CCN) activity in different ways, with the potential to have both direct and indirect impacts on atmospheric processes such as the formation and lifetime of clouds.

Given the relatively pristine Arctic environment, ship exhaust particle emissions could be a large perturbation to natural baseline Arctic aerosol concentrations. Low-level stratiform mixed-phase clouds cover large areas of the Arctic region and play an important role in the regional energy budget. Results from laboratory marine engine measurements, which investigated the impact of fuel sulfur content (FSC) reduction and wet scrubbing on exhaust particle properties, motivate the use of large eddy simulations to further investigate how such particles may influence the micro- and macrophysical properties of a stratiform

mixed-phase cloud case observed during the Arctic Summer Cloud Ocean Study campaign. Simulated enhancements of ship exhaust particles predominantly affected the liquid-phase properties of the cloud and led to a decrease in liquid surface precipitation, increased cloud albedo and increased longwave surface warming. The magnitude of the impact strongly depended on ship exhaust particle concentration, hygroscopicity, and size where the effect of particle size dominated the impact of hygroscopicity. While low FSC exhaust particles were mostly observed to affect cloud properties at exhaust particle concentrations

of 1000 $cm^{-3}$, exhaust wet scrubbing already led to significant changes at concentrations of 100 $cm^{-3}$. Additional simulations with cloud ice water path increased from $\approx 5.5$ $g\,m^{-2}$ to $\approx 9.3$ $g\,m^{-2}$, show more muted responses to ship exhaust perturbations but revealed that exhaust perturbations may even lead to a slight radiative cooling effect depending on the microphysical state





of the cloud. The regional impact of shipping activity on Arctic cloud properties may, therefore, strongly depend on ship fuel type, whether ships utilize wet scrubbers, and ambient thermodynamic conditions that determine prevailing cloud properties.

## 1    Introduction

Maritime shipping is a significant source of atmospheric pollutants with wide-ranging impacts on human health (Corbett et al., 2007; Jonson et al., 2020; Liu et al., 2016) and the climate system (Lauer et al., 2007; Eyring et al., 2010; Lund et al., 2012, 2020). Quantifying the net impact of ship exhaust emissions on Earth's radiative budget is a challenging task, due to large spatial variability in atmospheric conditions and heterogeneity in air exhaust composition. While ships emit a substantial amount of greenhouse gases, such as $CO_2$, the remaining constituents can vary substantially with the propulsion system and fuel type used by the individual vessel (Lack et al., 2009; Lack and Corbett, 2012; Lehtoranta et al., 2019). Whereas $CO_2$ emissions contribute to climate warming, the overall impact of particulate matter and $SO_2$ exhaust emissions is subject to a much larger uncertainty envelope.

Ship exhaust emissions of primary and secondary particles have been identified to lead to tens of thousands of premature deaths worldwide (Corbett et al., 2007). Regions that are particularly affected by these emissions include coastal areas and port cities, with high population densities, including parts of Europe and East Asia. Motivated by the harmful effects that ship exhaust particles have for human health, the International Maritime Organization (IMO) decided to introduce international marine fuel regulations, which primarily target a reduction of sulfur oxides ($SO_x$). These regulations mandate that ship operators use marine fuels with fuel sulfur content (FSC) lower than 0.5 wt % effective globally and lower than 0.1 wt % in designated sulfur emission control areas (SECA), or utilize exhaust treatment systems to reduce emissions (IMO, 2008). Since low FSC fuels are generally associated with a higher cost than conventional, high FSC residual fuel oils (UNCTAD, 2022) wet scrubbing systems pose an economically attractive treatment alternative, which allows stakeholders to continue to use marine fuels with FSCs exceeding regulatory limits (IMO, 2008). Wet scrubbers are exhaust after-treatment systems, that utilize mists of seawater or chemically treated freshwater to remove $SO_x$ from ships' exhaust, and thus, prevent the formation of sulfur-containing, secondary aerosol particles (Oikawa et al., 2003; Andreasen and Mayer, 2007). While recent studies demonstrate that utilization of FSC-lean marine fuels generally reduces the amount of particles emitted by ships (Zetterdahl et al., 2016; Kuittinen et al., 2021; Seppälä et al., 2021), the impact of wet scrubbing on particle exhaust emissions is less well understood and subjected to a large variability (Fridell and Salo, 2016; Lehtoranta et al., 2019; Winnes et al., 2020; Yang et al., 2021; Jeong et al., 2023). Moreover, compliance alternatives, such as exhaust after-treatment systems, have been found to affect the physicochemical properties of exhaust particles in different ways, which also have implications for atmospheric processes and the net climate effect of shipping activity. Combustion of low FSC fuels often results in the emission of predominantly hydrophobic soot particles, leading to reduced emissions of cloud condensation nuclei (CCN) compared to higher CCN emissions from conventional, high FSC fuel combustion (Lack et al., 2009; Yu et al., 2020, 2023). In contrast, wet scrubbing has been found to alter the physicochemical properties of the particle emissions (Lieke et al., 2013; Santos et al., 2023, 2024). This can yield larger fractions of water-soluble content in the exhaust particle phase and a shift in particle size distributions to larger



particles compared to exhaust particle emissions from conventional, high FSC fuel combustion. Combustion particles from wet scrubbing require relatively low supersaturations to be activated into liquid droplets (Santos et al., 2023) which can lead to enhanced CCN number emissions at given supersaturations (Santos et al., 2023, 2024).

Shipping emissions are currently estimated to have a net cooling effect on the climate; higher exhaust particle number concentrations can lead to increased cloud reflectivity which dominates the warming effect of shipping-related $CO_2$ emissions (Lauer et al., 2007; Eyring et al., 2010; Lund et al., 2012, 2020). Ship tracks are the visible manifestation of ship exhaust perturbations on cloud properties, resulting in persistent, regionally constrained marine stratiform cloud features with increased cloud albedo (Coakley et al., 1987; Hobbs et al., 2000; Possner et al., 2018). The extent of ship tracks depends on the background state of the boundary layer including meteorological parameters, the cloud fraction, and aerosol particle and CCN number concentrations (Coakley et al., 1987; Durkee et al., 2000; Hobbs et al., 2000). Observations along the coast of California have shown that the 0.1 wt % FSC limit, introduced in 2015 in SECAs, led to strong reductions in visible ship track formation (Gryspeerdt et al., 2019; Watson-Parris et al., 2022). While ship sulfate emissions are one key driver to ship track formation, FSC reduction policies may still lead to cloud perturbations. These cloud perturbations may be undetectable for some analysis techniques, resulting in an underestimate of shipping-induced radiative forcing (Gryspeerdt et al., 2019; Manshausen et al., 2022). With the introduction of the global 0.5 wt % FSC cap in 2020 and associated implications for exhaust particles, radiative cooling induced by ship exhaust emissions may be diminished. Studies investigating the impact of the 2020 0.5 wt % FSC cap have reported lower ship track formation frequencies and highlight the reduction in $SO_2$ emissions as key drivers for this observation (Gryspeerdt et al., 2019; Yuan et al., 2022; Watson-Parris et al., 2022). Therefore, IMO FSC regulations may imply a diminished radiative cooling from shipping emissions. However, the magnitude of diminished cooling may be subject to a systematic underestimate, as ship track visibility is strongly dependent on the clouds' background states (Gryspeerdt et al., 2019; Yuan et al., 2022; Watson-Parris et al., 2022).

One region where future shipping activity might lead to a strong climate feedback is the Arctic. The Arctic is experiencing unprecedented amplified surface warming compared to the global average, caused by a complex system of interacting processes within its climate system (Serreze and Francis, 2006; Serreze and Barry, 2011; Rantanen et al., 2022). Low-level mixed-phase clouds play a key role in the Arctic climate system (Morrison et al., 2012). Whereas low-level clouds generally lead to surface cooling, they tend to enhance surface warming in the Arctic throughout most of the year by trapping and re-emitting longwave radiation (Intrieri et al., 2002; Shupe and Intrieri, 2004). Enhanced surface warming in the Arctic promotes ice and snow melting and as a consequence, Arctic sea-ice extent and thickness have been in decline for the past decades (Screen and Simmonds, 2010; Serreze and Barry, 2011). This will likely grant ships easier access to exploration and extraction of natural resources and may enable the use of shorter trading routes through Arctic waterways, deviating from the more conventional and longer routes through the Suez and Panama Canals. The economic feasibility of Arctic shipping routes compared to traditional routes is debated (Lasserre and Pelletier, 2011). Nonetheless, shipping activity and related exhaust emissions are expected to increase significantly within the near future (Corbett et al., 2010; Paxian et al., 2010; Peters et al., 2011). In the Arctic, ambient particle number concentrations are relatively low compared to other regions of the Earth and thus, relatively small absolute increases in aerosol concentrations can substantially impact cloud formation and properties (Mauritsen et al., 2011; Bulatovic



et al., 2021). Ship emissions may therefore become a strong, localized aerosol source that could alter the properties of Arctic clouds and thereby the radiative budget.

Several studies have investigated the potential impacts of increased Arctic shipping activity on Arctic cloud properties (Christensen et al., 2014; Possner et al., 2017; Gilgen et al., 2018; Eirund et al., 2019). Ship aerosol emissions were observed
to generate a shift towards the ice phase, reducing precipitation and increasing cloud albedo (Christensen et al., 2014; Possner et al., 2017). Possner et al. (2017) observed a noteworthy rise in liquid water content (LWC) when ship-emitted CCN surpassed $1000\,\mathrm{cm}^{-3}$. However, results were inconclusive in determining whether ship emission-related changes were sufficient to impact Arctic warming rates (Christensen et al., 2014; Possner et al., 2017). Gilgen et al. (2018) modeled significant impacts on Arctic cloud properties from shipping when exaggerated future Arctic ship emission inventories were used, i.e., when Arctic shipping
emissions for 2050 were increased by a factor of 10. In contrast, Stephenson et al. (2018) investigated the total climate impact from trans-Arctic shipping and found an increase in total cloud fraction and cloud liquid water path (LWP) due to CCN-enhancements from ship emissions, diminishing Arctic warming rates and exerting cooling rates on the order of $1°$ C by the end of the 21st century. Eirund et al. (2019) highlight how underlying surfaces influence the properties of mixed-phase clouds and thus, the impact of additional CCN from ship exhaust emissions may be weakened or strengthened, depending on the ice
cover.

The aim of our study is to investigate how ship exhaust particle perturbations influence the microphysical structure of an Arctic mixed-phase cloud and thereby its climate effect. We elaborate on the differences in different ship exhausts based on laboratory results (Santos et al., 2022, 2023, 2024). In this study, we use large-eddy simulation (LES) to simulate a well-characterized mixed-phase stratocumulus cloud observed during the Arctic Summer Cloud Ocean Study (ASCOS) campaign
(Tjernström et al., 2012, 2014). We systematically perturb the aerosol concentrations in the model domain to explore the effect of different types of ship exhausts. Whereas previous studies investigating cloud perturbations caused by ship exhaust emissions, used simplistic representations of physicochemical properties of ship exhaust particles, herein, we utilize detailed exhaust particle information obtained from laboratory marine engine experiments where the impact of FSC reduction and exhaust wet scrubbing on ship exhaust particle properties was examined (Santos et al., 2022, 2023, 2024). The model is initially
run with an ambient background aerosol concentration only. Subsequent model simulations utilize several potential ship aerosol concentrations with different particle size distributions, densities, and hygroscopicities, mirroring the effects of FSC reduction and wet scrubbing. We evaluate the role of ship aerosol properties in affecting the cloud's LWP and ice water path (IWP), and the concentration of cloud droplets and raindrops. The results are used to calculate changes in surface precipitation, cloud drop effective radius, cloud albedo, and other cloud properties, which have implications for the radiative surface budget. Potential
Arctic climate feedbacks from increased shipping activity, in the context of the adaption of different fuel types and propulsion technologies by ships, are discussed.



## 2 Methods

### 2.1 Laboratory pre-study - Physicochemical properties of ship exhaust aerosol

The experimental results used in this study are based on a series of laboratory experiments that were performed between 2019
and 2022 in Gothenburg, Sweden. More details on the laboratory experiments can be found in Santos et al. (2022, 2023, 2024).
Engine experiments were performed using stationary, marine test-bed diesel engines, fuel types of varying sulfur content, a
laboratory wet scrubber, and a range of gas and aerosol instrumentation quantifying physicochemical properties of exhaust
particles. For the simulations with the MISU MIT Cloud and Aerosol (MIMICA) LES model, the following parameters are
needed as input to describe the aerosol perturbation: particle size distributions, particle effective densities, and hygroscopicities.
Particle size distributions were measured using Scanning Mobility Particle Sizers (SMPS). We describe average particle size
distributions using the count median diameter (CMD) and the geometric standard deviation ($\sigma_g$). Effective particle densities
($\rho_{\mathrm{eff}}$) were determined by coupled SMPS and Aerodynamic Aerosol Classifier (AAC) measurements and calculated following
Tavakoli and Olfert (2014) and Santos et al. (2022, 2024). Exhaust particle hygroscopicities ($\kappa$) were determined from size-
selected CCN measurements, using a CCN counter (CCNc; CCN-100, Droplet Measurement Technologies (Roberts and Nenes,
2005)) and parameterizations by Petters and Kreidenweis (2007).

From Santos et al. (2022, 2023) we use results from measurements with high FSC fuel (HiS; HGO in the respective studies),
one low FSC fuel (LoS; MGO in the respective studies), and seawater scrubbing experiments, performed in combination with
HiS fuel combustion (WS; SWS in the respective studies). For the LES experiments, the results were simplified by assigning
identical size distributions to HiS and LoS which did not display substantial differences during the respective measurement
campaigns. The particle size distributions used in the model are shown in Fig. 1 a. Other results, which were used as input
parameters, such as average case-dependent $\rho_{\mathrm{eff}}$ and $\kappa$ values are listed in Table 1 and discussed further in Sect. 2.3.

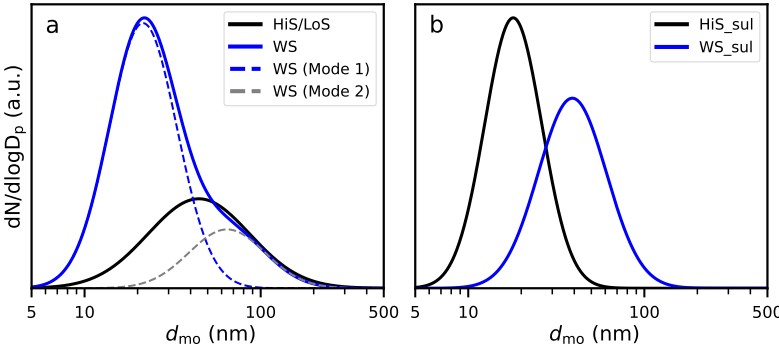

**Figure 1.** (a) Particle size distributions of high (HiS) and low sulfur content fuel (LoS), and wet scrubbed (WS) exhaust particles from
Santos et al. (2022, 2023) and (b) the sulfate particle modes of HiS fuel (HiS_sul) and scrubbed exhaust particles (WS_sul) from Santos et al.
(2024). The dashed lines in panel (a) represent the two individual modes of the bimodal WS case. The data shown in the figure represent
size distributions measured during the respective measurement campaigns that have been averaged and simplified to be parameterized within
MIMICA.




**Table 1.** Properties of marine background (BG) and ship aerosol used as model input parameters, including the count median diameter (CMD) and geometric standard deviation ($\sigma_g$) of the size distributions, particle density ($\rho_{\mathrm{eff}}$) and aerosol hygroscopicity ($\kappa$). HiS, LoS, and WS data were obtained from experiments outlined in Santos et al. (2022, 2023) and refer to combustion of high (HiS) and low FSC fuels (LoS), and wet scrubbed HiS exhaust (WS). HiS_sul and WS_sul represent the sulfate particle modes measured for high FSC fuel and wet scrubbed exhaust in Santos et al. (2024), respectively. The WS case is composed of a bimodal distribution, hence, the two separate aerosol modes are listed in the table. For each ship exhaust sensitivity test, two sets of simulations with either low or high ship aerosol number concentrations were performed. Corresponding simulations with low and high concentrations ($N_p$) are additionally labeled with _lo and _hi respectively.

| Case | CMD [nm] | $\sigma_g$ | $\rho_{\mathrm{eff}}$ [g cm$^{-3}$] | $\kappa$ | $N_p$ [cm$^{-3}$] |
|---|---|---|---|---|---|
| BG (Ait) | 32 | 1.1 | 2.18 | 1 | 30 |
| BG (Acc) | 93 | 1.5 | 2.18 | 1 | 30 |
| LoS | 45 | 1.6 | 0.91 | 0.04 | 100/1000 |
| HiS | 45 | 1.6 | 1.02 | 0.11 | 100/1000 |
| WS (Mode 1) | 22 | 1.2 | 1.18 | 0.22 | 131.3/1313 |
| WS (Mode 2) | 64 | 1.3 | 1.09 | 0.16 | 36.7/367 |
| HiS_sul | 18 | 1.15 | 1.6 | 0.64 | 100/1000 |
| WS_sul | 39 | 1.22 | 1.6 | 0.64 | 100/1000 |

From Santos et al. (2024), only results obtained for high FSC fuel combustion and exhaust wet scrubbing at engine load points of 50% were implemented in this study. Both cases resulted in emissions of bimodal size distributions consisting of a dominant, hygroscopic sulfate mode and a smaller, relatively hydrophobic soot mode. Here, corresponding bimodal particle size distributions were simplified to the respective unimodal, hygroscopic sulfate modes. The dominant sulfate modes (Fig.1 b) with their respective averaged $\rho_{\mathrm{eff}}$ and $\kappa$ values are summarized in Table 1. The high FSC case is referred to as HiS_sul and the seawater wet scrubbing case as WS_sul.

The main findings from Santos et al. (2022, 2023) showed that FSC reduction and exhaust wet scrubbing led to substantial impacts on particulate emissions from ship engines. A switch to marine fuels with reduced FSC did not significantly affect particle size distributions and total number emissions but decreased the exhaust particles' $\rho_{\mathrm{eff}}$ and $\kappa$ values. On the other hand, wet scrubbing was found to lead to the formation of a dominant particle mode around 20 nm, and to increased $\rho_{\mathrm{eff}}$ and $\kappa$ values, due to changes in the chemical mixing state. Similarly, Santos et al. (2024) investigated the impact of different fuel types and seawater exhaust wet scrubbing on exhaust particle properties, but used a different test-bed engine, with higher total power output. One key difference compared to Santos et al. (2022, 2023) was, that the combustion of non-compliant, high FSC fuel resulted in bimodal particle size distributions with a dominant sulfate mode around 20 nm. When the high FSC fuel exhaust was scrubbed, the sulfate mode was shifted towards larger sizes, likely due to the coagulation of particles inside the scrubber.



## 2.2 MIMICA model description and case setup

LES experiments were conducted with the MIMICA model. This model was originally designed to study high-latitude mixed-phase clouds and has been thoroughly documented and evaluated against observations (see e.g., Savre et al. (2015); Stevens et al. (2018); Bulatovic et al. (2023)). Herein, only a brief description of the model is provided. For more detailed information, see Savre et al. (2014).

MIMICA solves a set of anelastic, non-hydrostatic governing equations and uses a two-moment bulk microphysics scheme to predict mass mixing ratios ($Q$) and number densities ($N$) of five hydrometeor classes, including cloud droplets, raindrops, ice crystals, graupel, and snow. Growth of liquid-phase hydrometeors via auto-conversion, self-collection, and collision-coalescence are treated following Seifert and Beheng (2001) and Seifert and Beheng (2006). Interactions between liquid- and ice-phase hydrometeors are treated according to the two-moment bulk microphysics by Wang and Chang (1993). Hygroscopic growth of aerosol particles and activation into cloud droplets is calculated according to $\kappa$-Köhler theory (Petters and Kreidenweis, 2007). While MIMICA does include options for heterogeneous ice nucleation, here diagnostic ice crystal number concentrations ($N_i$) are utilized as in Ovchinnikov et al. (2011, 2014). This means that in grid cells where the temperature ($T$) is less than 0° C and sufficient supercooled cloud water is present ($Q_c \geq 2 \times 10^{-7}$ g m$^{-3}$), $N_i$ is relaxed towards a pre-determined, constant value. As a default, this value for $N_i$ was set to 200 m$^{-3}$ based on the control simulations in Stevens et al. (2018). The decision to use a constant diagnostic $N_i$ instead of an interactive heterogeneous ice nucleation scheme was motivated by findings showing that typical engine exhaust particles, such as black carbon, are inefficient ice nucleators in the immersion freezing regime (Mahrt et al., 2018; Kanji et al., 2020). In addition, Santos et al. (2024) found no significant differences in the ice nucleation behavior of exhaust particles emerging from low and high FSC fuel combustion, and exhaust wet scrubbing. The radiation solver used in this study is based on Fu and Liou (1992). It is important to note that while radiation is affected by cloud hydrometeors it is not affected affected by aerosols.

The simulated stratocumulus case is based on observations made during ASCOS on 31.08.2008 (see Appendix A with the detailed vertical profiles used to initialize the model; Tjernström et al., 2012, 2014). Note that meteolorogical conditions from ASCOS were only used utilized to initialize the model and remain constant for the remaining simulation time. Thus, they do not represent the temporal evolution of the real atmospheric state. The case study represents a stable mixed-phase cloud that has previously been investigated using MIMICA (Igel et al., 2017; Stevens et al., 2018; Christiansen et al., 2020; Sotiropoulou et al., 2021; Frostenberg et al., 2023). For more details on the setup of MIMICA please see the aforementioned references, and for more extensive information on the ASCOS campaign and the experimental results see Tjernström et al. (2012, 2014).

The MIMICA 3-D domain consists of $96 \times 96 \times 128$ grid cells with periodic boundaries. The horizontal resolution is uniform $dx = dy = 62.5$ m while the grid spacing in the vertical $z$-direction is variable 7.5 m$\leq dz \leq 25$ m. Higher vertical resolution is applied to grid cells near the surface and within the cloud layer, whereas a sinusoidal function is used to define the vertical spacing of grid cells at other altitudes. The total domain is 6 km $\times$ 6 km in the horizontal direction and 1.7 km in the vertical direction. All simulations were run for 16 h. The first 4 h are considered as spin-up and are thus excluded from the presented results.



Additional simulations of Mix, HiS_sul, and WS_sul were performed with $N_i$ increased from 200 to 600 m$^{-3}$. The aim of these additional simulations was to investigate the susceptibility of a thinner mixed-phase cloud, i.e., with reduced LWP and cloud depth, towards ship exhaust particle perturbations. A maximum value of $N_i = 600$ m$^{-3}$ as a large enough reduction in LWP was induced to perform aforementioned sensitivity tests and simultaneously, simulated LWP and IWP values agree well

with observational data (see Sect. 3.4). Additional testing revealed that further increases in $N_i$ would lead to dissipation of the cloud. Associated model runs are named as previous model runs but with an appended _ni600, e.g., Mix_ni600.

## 2.3   Aerosol implementation in MIMICA

Aerosol particles in MIMICA were represented as aerosol modes that follow lognormal distributions, described by a CMD and $\sigma_g$. To each aerosol mode, values of the aerosol effective density $\rho_{\text{eff}}$, the aerosol hygroscopicity expressed via $\kappa$, and the

aerosol number concentrations ($N_p$) were assigned. For all simulations, aerosol number concentrations and properties were set to be uniform and constant in time over the entire 3-D domain. Aerosol particles can be activated into cloud droplets according to $\kappa$-Köhler theory but are modeled without additional sources and sinks during the simulations. Several aerosol modes can co-exist. In our simulations, we describe the total aerosol by natural background aerosol modes and ship exhaust aerosol modes (for the cases with additional ship exhaust particles).

Natural background aerosol (BG) concentrations were present in all model runs (Table 1). These BG aerosols were assumed to have hygroscopicity values in agreement with marine seaspray ($\kappa = 1$) and were included in both the Aitken (Ait) and accumulation mode (Acc). The $N_p$ of the BG aerosol was chosen to be 30 cm$^{-3}$ in both modes based on aerosol measurements during the ASCOS expedition (Kupiszewski et al., 2013). In the baseline simulation (referred to as Mix), only BG Ait and Acc mode aerosol were present.

For the sensitivity experiments, different aerosol concentrations and types were added to represent different ship exhaust perturbations (HiS, LoS and WS from Santos et al. (2022, 2023), and HiS_sul and WS_sul from Santos et al. (2024)). Ship aerosol properties are summarized in Table 1 and Sect. 2.1. For each case, ship exhaust perturbation experiments were performed at two concentration levels, $N_{p,\text{ship}} = 100$ cm$^{-3}$ (labeled with _lo) and 1000 cm$^{-3}$ (labeled with _hi), respectively. An exception is the WS case where the concentration levels are increased by a factor of $\approx 1.7$ following the increase in particle

number concentration that has been observed in the experiments when using the wet scrubber. This increase in $N_{p,\text{ship}}$ was accounted for in the two particle modes comprising the WS case. The same _lo and _hi labeling, signifying the low and high concentration simulations, was used for WS cases.

## 2.4   Calculations of the cloud drop effective radius and cloud albedo

To examine the difference in cloud radiative properties between the simulations, we calculate the effective droplet radius ($r_e$),

cloud optical depth ($\tau$), and cloud albedo ($\alpha$) from the model output. To calculate the effective cloud droplet radius $r_v$ we use the relationship as suggested by Freud and Rosenfeld (2012), who found that $r_e$ is on average a factor 1.08 larger than the



volume mean cloud droplet radius $r_v$,

$$r_e \approx 1.08 \, r_v \ . \tag{1}$$

The volume mean cloud droplet radius $r_v$ is defined as,

$$r_v = \left( \frac{3}{4} \frac{Q_c}{\pi \rho_w N_c} \right)^{1/3} , \tag{2}$$

where $Q_c$ is the cloud liquid water content, $\rho_w$ is the density of water (1000 kg m$^{-3}$) and $N_c$ is the cloud droplet number concentration. The cloud's optical depth can be approximated by

$$\tau = \frac{3}{2} \frac{\mathrm{LWP}}{r_e \rho_w} , \tag{3}$$

where LWP is the liquid water path, i.e., the vertically integrated amount of liquid cloud water, in kg m$^{-2}$ (Stephens, 1978). The cloud albedo can be approximated with

$$\alpha = \frac{(1-g)\tau}{1+(1-g)\tau} , \tag{4}$$

where $g$ is the scattering asymmetry factor, i.e., the average value of the cosine of the scattering angle, and equals 0.85 for the scattering of solar radiation by clouds (Meador and Weaver, 1980).

## 3 Results

### 3.1 Influence of ship aerosol on LWP and IWP

In Fig. 2, the time evolution of the domain-averaged LWP and IWP are shown for all the simulations in comparison to the observation from the ASCOS campaign. Note that ASCOS observations used to initialize the model do not change with time during the remaining simulation period. In all simulations, MIMICA simulates an LWP that exceeds the 75th percentile of the observations (Fig. 2 a - b) which was also observed by Bulatovic et al. (2021), who used MIMICA to simulate the same ASCOS case. MIMICA was previously reported to generate greater LWP when prescribed instead of interactive aerosol particle concentrations are used (Stevens et al., 2018). Additional sensitivity tests with reduced LWP are discussed in Sect. 3.4. The addition of ship aerosol tends to increase the LWP of the cloud compared to the reference Mix case (Fig. 2 a - b; Table 2). This effect is found to be dependent on the ship exhaust aerosol concentrations and the hygroscopicity of the ship exhaust aerosol. The LWP increase is most pronounced for sensitivity tests with high ship aerosol concentrations (HiS_hi, WS_hi, and WS_sul_hi), where LWP increases by up to $\approx$13%. The increase is less pronounced for both LoS cases due to the hydrophobic nature of the added particles ($\kappa = 0.04$). Despite having comparatively large $\kappa$-values, both HiS_sul cases do not yield any substantial increase in LWP, suggesting that the ship exhaust aerosol are too small (CMD = 18 nm) to induce a pronounced effect. The identified LWP response for the mixed-phase cloud perturbed by ship exhaust agrees with Possner et al. (2017), who reported substantial increases in LWP when ship-related CCN concentrations exceeded 1000 cm$^{-3}$ in their simulations.



250    The simulated IWP is close to the 25th percentile of the observations for all simulations (Fig. 2 c - d). In contrast to results for the LWP, additional ship exhaust particles have no substantial effect on the modeled IWP which is due to the implementation of diagnostic $N_i$, meaning ship aerosol can not directly impact $N_i$. However, in our simulations, ship exhaust aerosol can affect the properties of precipitating ice-phase hydrometeors (graupel and snow) by influencing the accretion of hydrometeors and the availability of water vapor. Small increases in IWP for some simulations are mainly caused by an increasing graupel number

255    ($N_g$) and mass concentration ($Q_g$; Fig. C1).

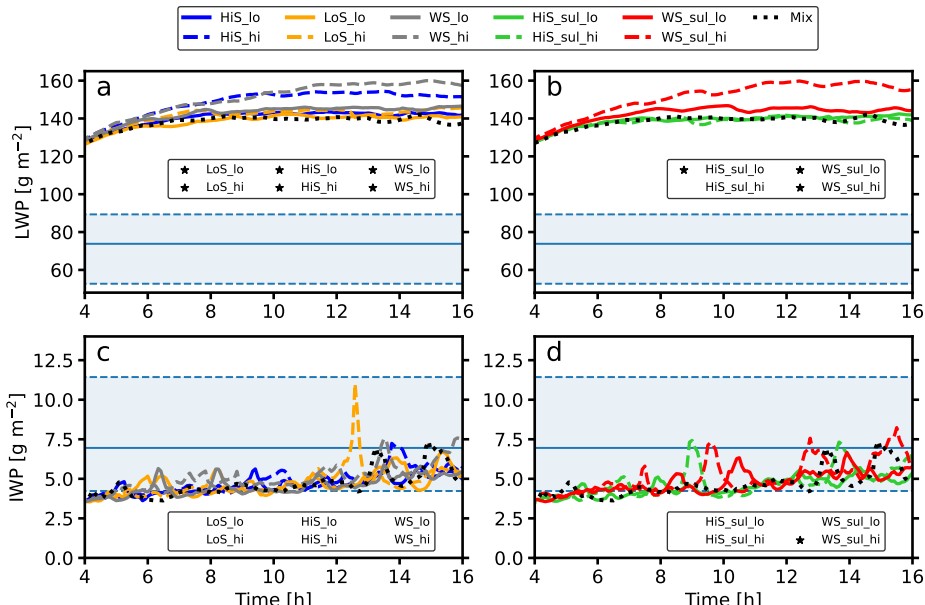

**Figure 2.** Time evolution of the simulated domain-averaged (a and b) liquid water path (LWP) and (c and d) ice water path (IWP). Mix refers to the reference case with background aerosol only. HiS, LoS, and WS represent ship aerosol from measurements of high and low sulfur content fuels and wet scrubbing respectively (Santos et al., 2022, 2023). The HiS_sul and WS_sul cases represent sulfate particle modes of high FSC fuel combustion and exhaust gas wet scrubbing from Santos et al. (2024). The label additions *_lo* and *_hi* signify the ship aerosol concentrations used in the individual model runs. Significant differences between ship exhaust cases and Mix were assessed using two-sided $t$ tests at a confidence level of 95%. Model runs with significant differences are marked with star icons in inset legends. For the statistical tests the last 4 simulation hours were used. The blue shaded area refers to the retrieved LWP and IWP from microwave radiometer measurements (median over the observation period; the corresponding dashed lines are the 25th/75th percentiles) during the ASCOS campaign (Tjernström et al., 2012, 2014) which were used to initialize the model. The first four hours are considered a spin-up period of the model and are removed from the figures.

## 3.2    Impact of ship aerosol on hydrometeors

The effect of ship exhaust aerosol perturbations on the temporal evolution of cloud hydrometeors and cloud depth is investigated by examining horizontally averaged number and mass concentrations of cloud droplets ($N_c$ and $Q_c$) and raindrops ($N_r$ and





**Table 2.** Overview of mean LWP, IWP, surface precipitation rates, cloud effective radius ($r_e$), cloud albedo ($\alpha$), and net long- (Net LW) and shortwave radiation at the surface (Net SW) averaged over the last four hours of simulation time. Statistical significance, determined by performing two-side t-tests, is highlighted by asterisks. The label additions _lo and _hi signify the ship aerosol concentrations used in the individual model runs.

| Case | LWP [g m$^{-2}$] | IWP [g m$^{-2}$] | Surface precip. (Total) [mm d$^{-1}$] | Surface precip. (Rain) [mm d$^{-1}$] | $r_e$ [$\mu$m] | $\alpha$ | Net LW [W m$^{-2}$] | Net SW [W m$^{-2}$] |
|---|---|---|---|---|---|---|---|---|
| Mix | 139.3 | 5.5 | 0.32 | 0.06 | 16.56 | 0.65 | -14.6 | 6.5 |
| LoS_lo | 140.7* | 5.3 | 0.31 | 0.05 | 16.50 | 0.65* | -14.5 | 6.5 |
| LoS_hi | 143.6* | 5.5 | 0.30 | 0.04* | 16.16* | 0.66* | -14.4* | 6.5 |
| HiS_lo | 142.5* | 5.1 | 0.31 | 0.06 | 16.41* | 0.66* | -14.5 | 6.5 |
| HiS_hi | 152.8* | 5.6 | 0.28* | 0.03* | 15.41* | 0.68* | -14.2* | 6.4 |
| WS_lo | 145.4* | 5.1 | 0.30 | 0.06 | 16.22* | 0.66* | -14.4* | 6.5 |
| WS_hi | 158.6* | 5.9 | 0.27* | 0.02* | 14.68* | 0.70* | -13.9* | 6.4 |
| HiS_sul_lo | 141.0* | 5.0 | 0.31 | 0.06* | 16.57 | 0.65* | -14.6 | 6.5 |
| HiS_sul_hi | 139.8 | 5.4 | 0.31 | 0.06 | 16.52 | 0.65 | -14.6 | 6.5 |
| WS_sul_lo | 144.6* | 5.4 | 0.30 | 0.05* | 16.10* | 0.67* | -14.4* | 6.5 |
| WS_sul_hi | 158.0* | 6.2* | 0.29 | 0.02* | 14.69* | 0.70* | -14.0* | 6.3 |
| Mix_ni600 | 88.6 | 9.3 | 0.35 | 0.00 | 15.51 | 0.57 | -17.6 | 6.9 |
| HiS_sul_lo_ni600 | 86.6* | 9.4 | 0.36 | 0.00 | 15.47* | 0.57* | -17.9* | 6.9 |
| HiS_sul_hi_ni600 | 85.8* | 9.4 | 0.35 | 0.00 | 15.46* | 0.57* | -18.0* | 6.9 |
| WS_sul_lo_ni600 | 88.0* | 9.2 | 0.34 | 0.00 | 14.94* | 0.58* | -17.9* | 6.9 |
| WS_sul_hi_ni600 | 86.8* | 9.6 | 0.35 | 0.00 | 13.59* | 0.60* | -17.9* | 6.9 |

$Q_r$). Corresponding contour plots for the reference simulation (Mix) and all sensitivity simulations with high ship aerosol
concentrations (_hi; 1000 cm$^{-3}$ or 1680 cm$^{-3}$ (WS)) are shown in Fig. 3. In addition, each subplot features the horizontally averaged cloud bottom and cloud top height, depicted by the black dashed lines.

In general, impacts of ship aerosol particles on liquid-phase hydrometeors are mostly observed in ship aerosol simulations with the highest $N_{p,\text{ship}}$ in accordance with the results presented in Sect. 3.1. Ice-phase hydrometeors have been excluded from Fig. 3, as changes in ice crystal concentrations remain more or less unaffected due to the prescribed ice parameterization scheme used in this study (see Sect. 2.2), but can be found in Appendix B. Whereas cloud droplet numbers are evenly distributed within the cloud layer, $Q_c$ is highest near the cloud top, which is typical for stratocumulus clouds with near-adiabatic conditions. The distribution of $N_r$ displays a more dynamic behavior and reaches its maximum after around 8 hours of simulation. In contrast to cloud droplets, $Q_r$ is concentrated towards the lower regions of the cloud. All simulations show similar cloud depths and evolutions (Fig. 3). After 4 h (spin-up) the clouds have a depth of around 500 m. The cloud depth increases in all simulations





270 and ranges between 535 m (Mix) and 570 m (WS_sul_hi) at the end of the simulations due to steady increases in cloud top height.

**Figure 3.** Temporal evolution of horizontal domain averaged cloud droplet number concentrations ($N_c$; $[N_c] = \text{m}^{-3}$), cloud droplet mixing ratios ($Q_c$; $[Q_c] = \text{g m}^{-3}$), raindrop number concentrations ($N_r$) and raindrop mixing ratios ($Q_r$) simulated for the reference case (Mix) and the high ship aerosol concentration cases LoS_hi, HiS_hi, WS_hi, HiS_sul_hi and WS_sul_hi. The black dashed lines represent case-specific, horizontally averaged cloud bottom and cloud top heights. The spin-up period (0 to 4 h) is removed from all figures.



With the addition of ship aerosol, more aerosol particles are activated into cloud droplets as can be seen from increased $N_c$ values for LoS_hi, HiS_hi, WS_hi, HiS_sul_hi and WS_hi (Fig. 3 and Fig. 4). The largest increase is observed for WS_hi and WS_sul_hi where the vertically integrated $N_c$ averaged over the last 4 simulation hours increases by $\approx 57\%$. Note that $Q_c$ is almost unaffected by the added ship exhaust aerosol due to the low precipitation rates in the Mix case. Despite the relatively large $\kappa$ value of HiS_sul_hi ship exhaust particles ($\kappa = 0.64$), $N_c$ and $N_r$ are not strongly affected, even when ship exhaust aerosol concentrations are set to $N_p = 1000\,\mathrm{cm}^{-3}$. This implies that HiS_sul exhaust particles were too small (CMD = 18 nm) to act as CCN. The observed increase in $N_c$ is also observed for LoS_hi, suggesting that additional aerosol particles of low hygroscopicity ($\kappa = 0.04$) can impact cloud properties, given their CMD is sufficiently large. Differences in modeled $N_c$ between HiS_sul_hi and LoS_hi ship exhaust aerosol imply that the size of aerosol particles plays a more dominant role in inducing changes in cloud properties than particle hygroscopicity. This observation agrees with Christiansen et al. (2020), who simulated the same ASCOS case with background aerosol modes of varying size and hygroscopicity. Therein, the authors found microphysical cloud properties were not affected by aerosol particles' hygroscopicities if accumulation mode particles were present in the model domain (Christiansen et al., 2020).

Vertical profiles of $N_c$, $Q_c$, $N_r$, and $Q_r$ averaged over the last four simulation hours reveal a more detailed picture of how ship perturbations affect concentrations of cloud droplets and raindrops (Fig.4). The sensitivity tests that show substantial increases in $N_c$ also show reduced raindrop formation in the cloud (Fig. 3 and Fig. 4). Whereas Mix produces substantial amounts of raindrops near the cloud top after about 6 to 7 h of simulation, ship cases with high exhaust particle concentrations lead to general reductions in $N_r$ and $Q_r$ by up to 58% and 63% respectively (Fig. 3 and Fig. 4). The magnitude of this response is dependent on the CMD and $\kappa$, where the CMD effect dominates the cloud response. The strongest reduction in $N_r$ and $Q_r$ is observed for WS_hi and WS_sul_hi, where both quantities are reduced by about 52 to 58% ($N_r$) and 56 to 63% ($Q_r$) compared to Mix. Results for raindrop formation coincide with a general reduction in $r_e$ for relevant ship exhaust cases. For both WS_hi and WS_sul_hi, $r_e$ is reduced from 16.56 $\mu$m (Mix) to $\approx 14.68$ $\mu$m (Fig. 5 and Table 2). A reduction in $r_e$ indicates reduced self-collection (autoconversion) and coalescence/accretion of cloud droplets by raindrops or other hydrometeors.

## 3.3 Impact of ship aerosol on surface precipitation and cloud radiative properties

The sensitivity tests that led to reductions in $N_r$ and $Q_r$ are not found to impact total surface precipitation rates compared to Mix (Fig. 5 a - b). The majority of surface precipitation is dominated by graupel which is not found to be affected by the addition of ship exhaust aerosol. Liquid rain typically constitutes less than 5% of the total surface precipitation (Table 2). Despite the relatively low absolute rates, rain surface precipitation rates are found to be reduced with additional ship exhaust aerosol, agreeing with tendencies in ship exhaust cases to produce smaller $r_e$ (Fig. 5 c - d and Table 2). Changes in surface precipitation rates may significantly change with more realistic ice formation parametrizations. As a result, we cannot exclude whether emissions associated with shipping activity may extend cloud lifetimes due to potential reductions in total precipitation rates (Albrecht, 1989).

In order to estimate the potential climatic impact of increased Arctic shipping activity, $\alpha$, net short- (SW) and longwave (LW) radiative fluxes at the surface are characterized (Fig. 5 e - f and Table 2). Net radiative fluxes are calculated by subtracting net



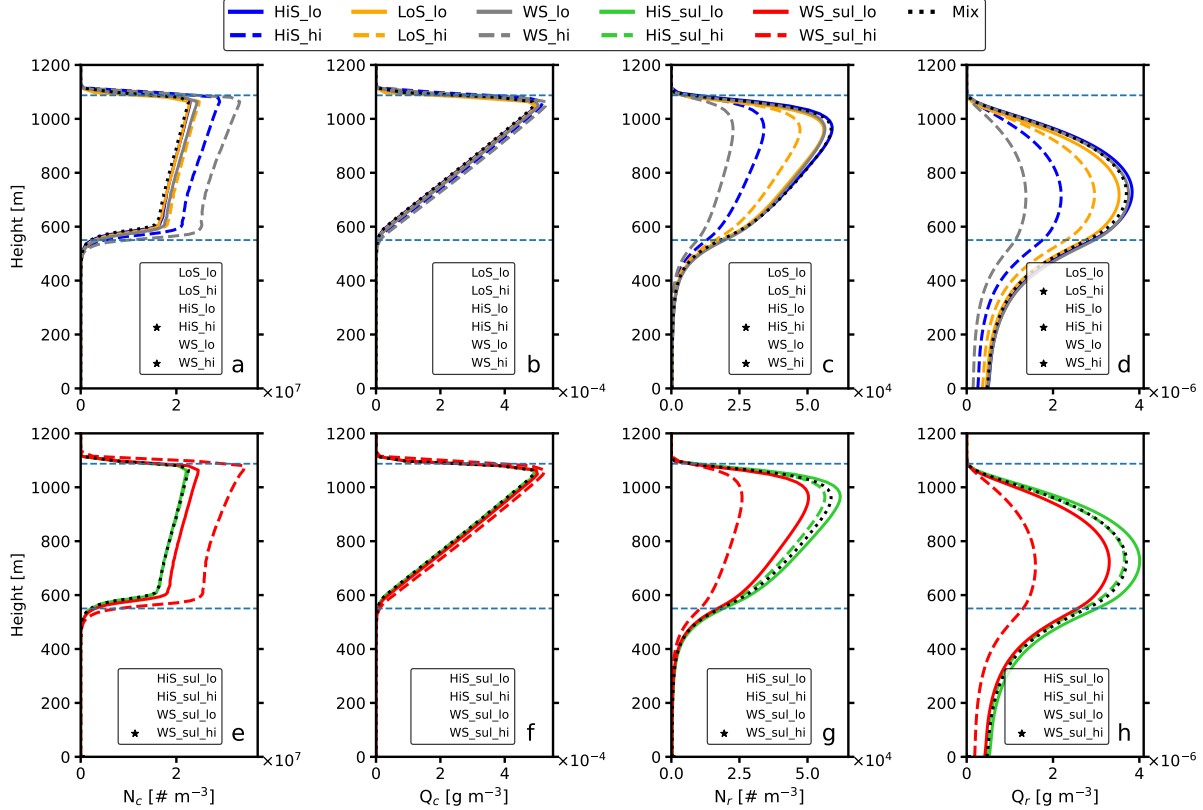

**Figure 4.** Vertical profiles of horizontally averaged (a and e) $N_c$, (b and f) $Q_c$, (c and g) $N_r$ and (d and h) $Q_r$ averaged over the last four simulation hours. The light blue, dashed line represents the average cloud bottom and top height calculated for the reference case (Mix). HiS, LoS, and WS represent ship aerosol from measurements of high and low sulfur content fuels and wet scrubbing respectively (Santos et al., 2022, 2023). The HiS_sul and WS_sul cases represent sulfate particle modes of high FSC fuel combustion and exhaust gas wet scrubbing from Santos et al. (2024). The label additions _lo_ and _hi_ signify the ship aerosol concentrations used in the individual model runs. Significant differences between ship exhaust cases and Mix were assessed using two-sided $t$ tests at a confidence level of 95%. Model runs with significant differences are marked with star icons in inset legends.

upwelling fluxes from net downwelling fluxes, hence, a negative value implies net outgoing radiation. At high latitudes, LW radiation generally has a larger influence on the surface energy budget compared to SW radiation, as solar radiation is limited outside the summer months. However, since the ASCOS case used in this study is based on observations from August, the net SW at the surface is also investigated.

With the exception of HiS_sul_hi, all ship sensitivity simulations tend to significantly increase $\alpha$ compared to Mix (Fig. 5 e - f). This observation agrees with results shown in Fig. 4 and Fig. 5 c - d where tendencies towards generating larger $N_c$ and reduced $r_e$ values are shown. The largest increase in $\alpha$ is observed for WS_hi and WS_sul_hi where cloud albedo increases





from 0.65 to 0.70. The changes in LWP and $r_e$ induced by ship aerosol perturbations are also seen in the LW net radiative fluxes at the surface (Fig. 5 g - h).

Net LW fluxes at the surface are negative, meaning the net radiative LW flux is upwelling and therefore, cooling the surface. After about 5 h of simulation, the net LW surface fluxes reach their respective radiative cooling maxima of $\approx$-13 W m$^{-2}$ and eventually decrease to $\approx$-15 W m$^{-2}$ (Mix) and $\approx$-14.3 W m$^{-2}$ (WS_hi) towards the end of the simulation. Ship cases, which are found to lead to the largest increase in LWP (His_hi, WS_hi, and WS_sul_hi; Fig. 2 a - b), also reduce net LW cooling at the surface compared to Mix, i.e., net LW becomes less negative. Our results suggest that ship exhaust perturbations may lead

to diminished surface LW radiative cooling and could therefore lead to enhanced surface warming, that is if concentrations and the CMD of the associated exhaust particle size distributions are sufficiently large to act as CCN. Similar relationships between increased LWP and a reduced LW radiative cooling, was also noted by Christiansen et al. (2020).

The net SW radiation is positive in all simulations, meaning the net flux is downwelling. In all simulations, the net SW fluxes initially increase until 6 h into the respective simulations where a maximum of around 14 W m$^{-2}$ is reached. By the end of the

simulations, net SW decreases to $\approx$5 W m$^{-2}$. The temporal trends in LW and SW radiation both coincide with the solar angle. However, none of the ship sensitivity tests are found to significantly impact net SW fluxes at the surface, despite associated increases in $\alpha$ (Fig. 5 i - j), which is expected given the relatively large LWP (Fig. 2 and Table 2).

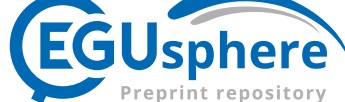

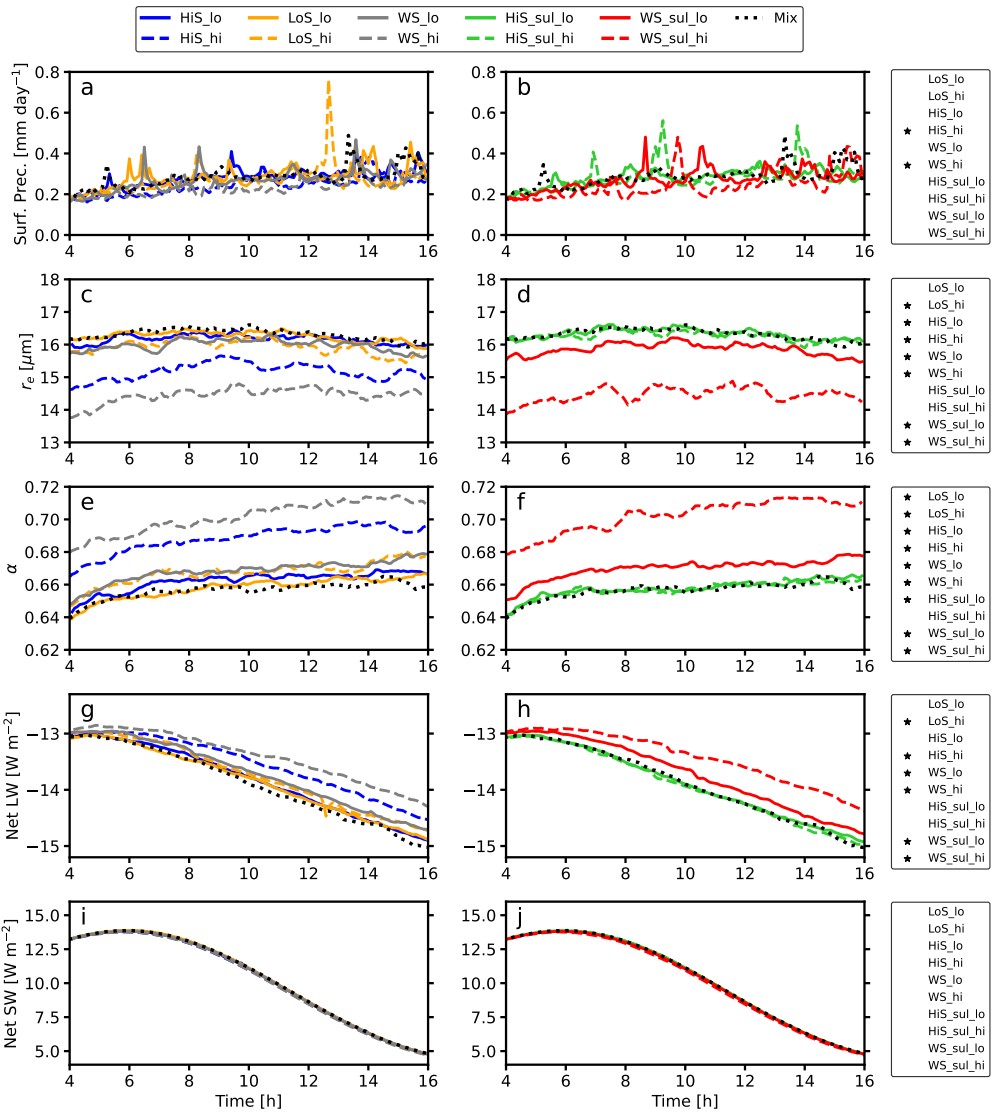

**Figure 5.** Time evolution of the simulated domain-averaged (a and b) surface precipitation, (c and d) $r_e$, (e and f) $\alpha$, (g and h) net longwave radiation at the surface (Net LW) and (i and j) net shortwave radiation at the surface (Net SW) for the set of simulations. Net radiative fluxes are calculated by subtracting the upwelling radiative flux from the downwelling flux (e.g., $LW_{down}-LW_{up}$), hence, a negative value implies net outgoing radiation. Mix refers to the reference case with background aerosol only. HiS, LoS, and WS represent ship aerosol from measurements of high and low sulfur content fuels and wet scrubbing respectively (Santos et al., 2022, 2023). The HiS_sul and WS_sul cases represent sulfate particle modes of high FSC fuel combustion and exhaust gas wet scrubbing from Santos et al. (2024). The label additions _lo and _hi signify the ship aerosol concentrations used in the individual model runs. Significant differences between ship exhaust cases and Mix were assessed using two-sided $t$ tests at a confidence level of 95%. Model runs with significant differences are marked with star icons in right-hand side legends. The last 4 simulation hours were used to perform statistical tests.





### 3.4 Sensitivity to different levels of $N_i$

Additional simulations of Mix, HiS_sul, and WS_sul were performed with $N_i$ increased from 200 to 600 m$^{-3}$. The aim of these

additional simulations was to investigate the susceptibility of a thinner mixed-phase cloud, i.e., with reduced LWP and cloud depth, towards ship exhaust particle perturbations. When $N_i$ is increased to 600 m$^{-3}$, the LWP is reduced by $\approx$45% compared to Mix and is close to the 75th percentile of the observational data (Fig. 6 a). IWP is increased by 3.8 g m$^{-2}$ ($\approx$69%; comparison between Mix and Mix_ni600) and is also within the observed range (Fig. 6 b). With an increased $N_i$, ship exhaust particles do not lead to as strong perturbations in LWP ($|\Delta$LWP$| < 3.2\%$) compared to sensitivity tests performed with $N_i = 200$ m$^{-3}$. In

fact, the ship exhaust sensitivity simulations display a tendency towards reducing the LWP compared to Mix_ni600 which is in contrast to the model runs with $N_i = 200$ m$^{-3}$ shown in Fig. 2. Such a muted response in LWP adjustments from additional ship exhaust particles was also reported by Possner et al. (2017) who found a suppressed LWP response from ship-related CCN emissions when the number of ice nucleating particles was increased. The IWP, on the other hand, is increased for Mix_ni600, HiS_sul_hi_ni600 and WS_sul_hi_ni600 compared to Mix (Fig. 6 b) as all ice-phase hydrometeors increase substantially in

number and mass concentrations (Fig. 7 c - e and h - j). As with the first set of simulations, ship exhaust perturbations do not significantly impact the IWP compared to the respective reference case, Mix_ni600 which, as previously discussed, is mainly due to the implementation of diagnostic $N_i$.

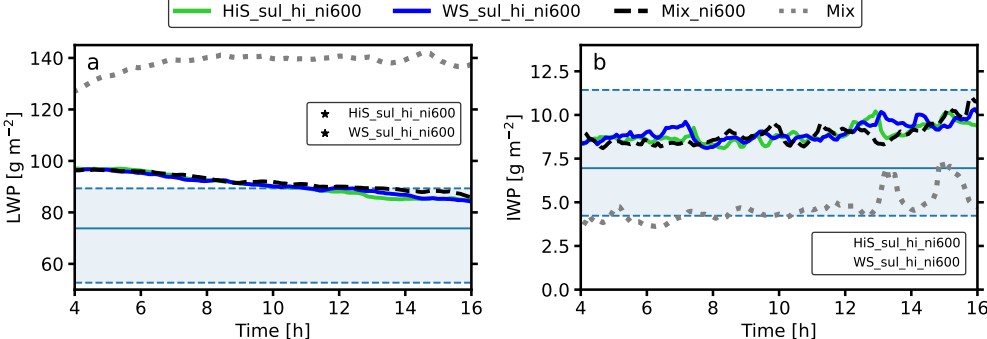

**Figure 6.** Time evolution of the simulated domain-averaged (a) liquid water path (LWP) and (b) ice water path (IWP). Figures show results for model runs where $N_i$ was increased from 200 to 600 m$^{-3}$. Model runs with increased $N_i$ are labeled with _ni600. The simulations are compared to the Mix case where $N_i = 200$ m$^{-3}$. The blue shaded area refers to the retrieved LWP and IWP from microwave radiometer measurements (median over the observation period; the corresponding dashed lines are the 25th/75th percentiles) during the ASCOS campaign (Tjernström et al., 2012, 2014). Only ship exhaust sensitivity cases with high concentrations (_hi) are shown in the figure. Significant differences between ship exhaust cases and Mix_ni600 were assessed using two-sided $t$ tests at a confidence level of 95%. Model runs with significant differences are marked with star icons in inset legends. The last 4 simulation hours were used to perform statistical tests. The first 4 hours are considered a spin-up period of the model and are removed from the figures.

Fig. 7 shows the impact on cloud depth and all hydrometeor classes for the _ni600 simulations compared to Mix (with $N_i = 200$ m$^{-3}$). The cloud depth is decreased, mostly due to adjustments in cloud bottom height, which increases by $\approx$80 m





(Fig. 7). Taking into account the increase in cloud bottom height, $N_c$ values are either similar (Mix_ni600, HiS_sul_hi_ni600) or increased (WS_sul_hi_ni600) compared to the respective Mix values (Fig. 7 a). Simultaneously, $Q_c$ values of all _ni600 cases are reduced compared to Mix, suggesting smaller cloud droplets in the cloud (Fig. 7 f). Raindrop number $N_r$ and mass concentrations $Q_r$ are significantly reduced compared to Mix (Fig. 7 b - g), resulting from reduced auto-conversion of cloud droplets to raindrops and droplet coalescence efficiency, as well as enhanced scavenging of raindrops by graupel and snow,

which display substantial concentration increases in all _ni600 runs (Fig. 7 d - e and Fig. 7 i - j).

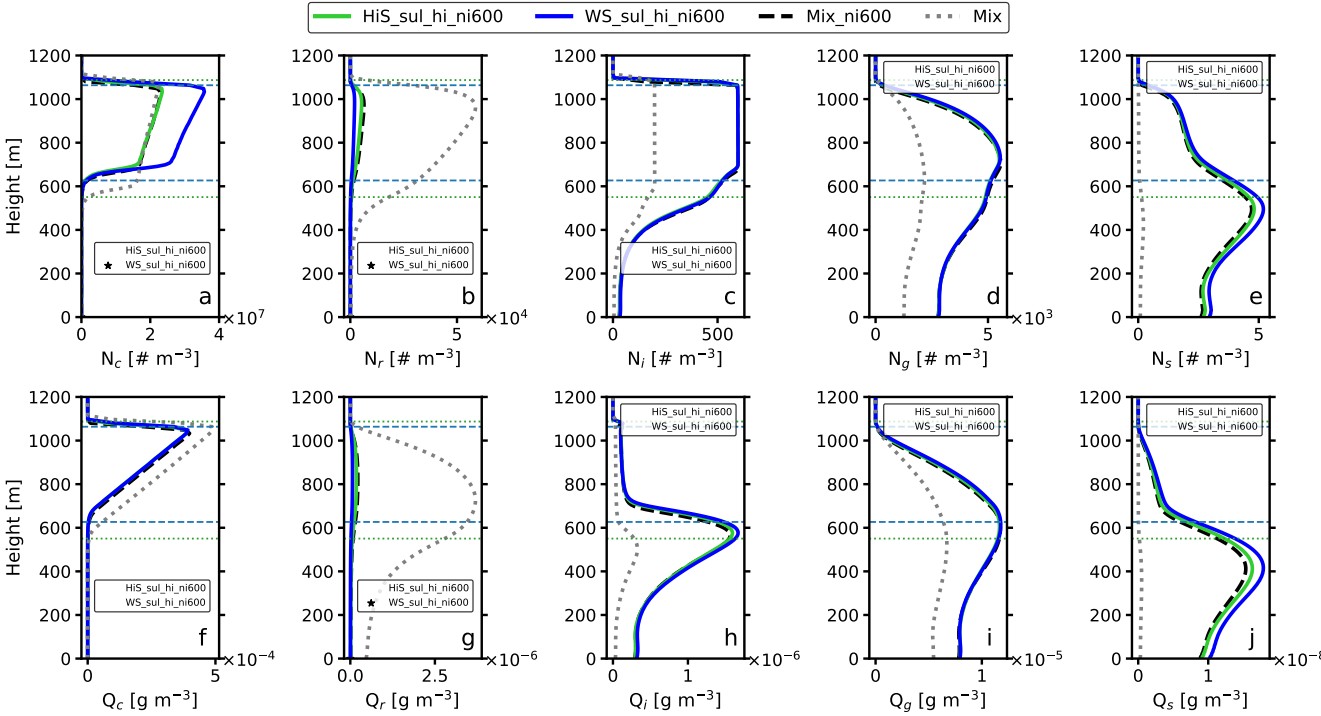

**Figure 7.** Vertical profiles of horizontally averaged (a) $N_c$, (b) $N_r$, (c) $N_i$, (d) $N_g$, (e) $N_s$, (f) $Q_c$, (g) $Q_r$, (h) $Q_i$, (i) $Q_g$ and (j) $Q_s$ averaged over the last four simulation hours for Mix_ni600, HiS_sul_hi_ni600, WS_sul_hi_ni600 and Mix. The light blue, dashed line represents the average cloud bottom and top height calculated for Mix_ni600 (green, dotted line for Mix respectively). The HiS_sul and WS_sul cases represent sulfate particle modes of high FSC fuel combustion and exhaust gas wet scrubbing. Significant differences between ship exhaust cases and Mix_ni600 were assessed using two-sided $t$ tests at a confidence level of 95%. Model runs with significant differences are marked with star icons in inset legends.

Surface precipitation rates are slightly increased in the _ni600 simulations compared to Mix ($< 10\%$; Table 2 and Fig. 8 a) which is due to enhancements in graupel formation (Fig. 7 d and i). Liquid-phase surface precipitation rates are small and negligible for all _ni600 simulations. Average cloud drop effective radii $r_e$ values are reduced by $\approx 1\ \mu$m compared to the respective simulations with $N_i = 200\ \text{m}^{-3}$ agreeing with observed reductions in LWP (Fig. 6 a) and $Q_c$ (Fig. 7 f). The reductions in LWP

and $\alpha$ are both reflected in the net radiative fluxes at the surface. Net LW radiative fluxes at the surface are reduced by at least





2 W m$^{-2}$ compared to Mix, meaning that reductions in LWP and cloud depth lead to reduced re-emission of LW radiation to the surface (Fig. 8 d). Respective _ni600 ship exhaust cases tend to reduce net LW at the surface further, suggesting that in these instances ship exhaust perturbations would yield a slight net cooling effect. This is in contrast to the results shown in Sect. 3.3 where ship exhaust perturbations reduced net outgoing LW radiation at the surface. Relative reductions in $\alpha$ for _ni600 cases

are reflected in net SW fluxes at the surface. In comparison to Mix, net SW is on average increased by $\approx$0.5 W m$^{-2}$ compared to Mix (Fig. 8 e and Table 2). Nevertheless, additional ship exhaust particles do not significantly alter net SW radiative surface fluxes.

In summary, the results shown in this section demonstrate that the impact of ship exhaust particles on clouds and the surface radiative budget does not only depend on the ship exhaust particles themselves. The sensitivity is also strongly dependent on the

background state of the atmosphere and the background cloud properties, such as cloud thickness. Interestingly, ship exhaust perturbations can have opposite effects on certain cloud parameters and net radiative fluxes. The most striking difference is that ship emissions tend to decrease the LWP and enhance net outgoing LW radiative surface fluxes when $N_i = 600$ m$^{-3}$, whereas the opposite is obtained when $N_i = 200$ m$^{-3}$.



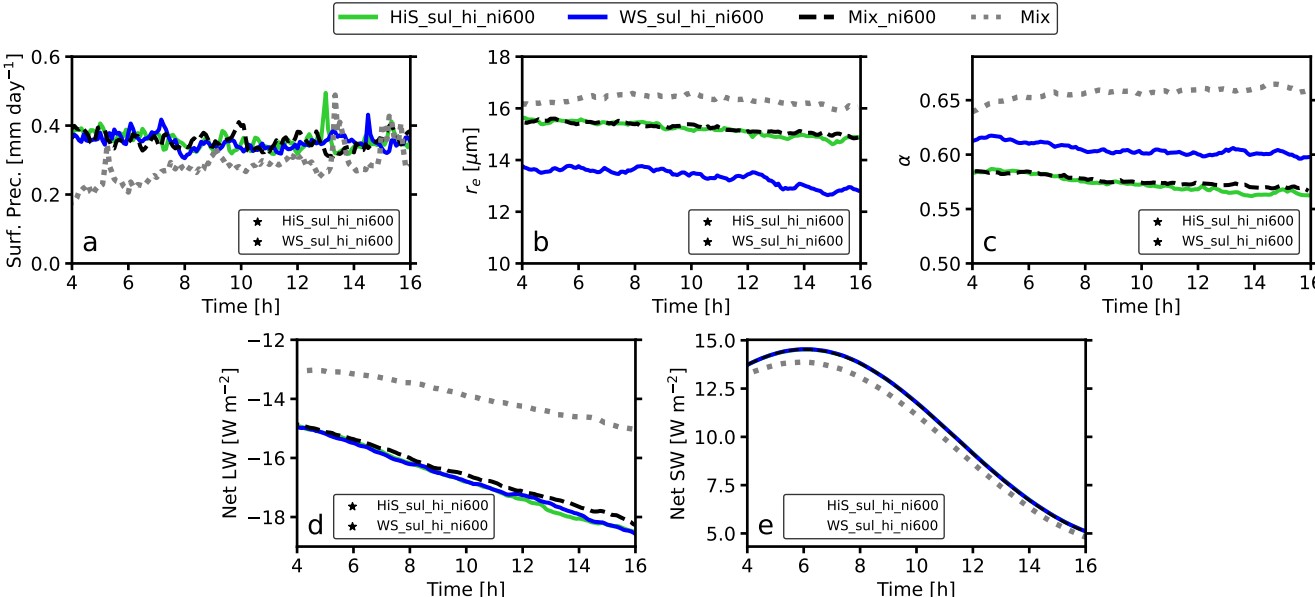

**Figure 8.** Time evolution of the simulated domain-averaged (a) surface precipitation rates, (b) $r_e$, (c) $\alpha$, (d) net longwave radiation at the surface (Net LW) and (e) net shortwave radiation at the surface (Net SW) for a set of simulations with $N_i = 600$ m$^{-3}$ (_ni600) and the Mix case ($N_i = 200$ m$^{-3}$). Net radiative fluxes are calculated by subtracting the upwelling radiative flux from the downwelling flux (e.g., LW$_{down}$-LW$_{up}$), hence, a negative value implies net outgoing radiation. Mix refers to the reference case with background aerosol only. The HiS_sul and WS_sul cases represent sulfate particle modes of high FSC fuel combustion and exhaust gas wet scrubbing from Santos et al. (2024). Only ship exhaust sensitivity cases with high concentrations (_hi) are shown in the figure. Significant differences between ship exhaust cases and Mix_ni600 were assessed using two-sided $t$ tests at a confidence level of 95%. Model runs with significant differences are marked with star icons in inset legends. The last 4 simulation hours were used to perform statistical tests.



## 4 Discussion

In general, ship emissions can lead to more but smaller liquid droplets in the mixed-phase clouds studied here. This means that even if the clouds contain more liquid water, rain surface precipitation is reduced. Nevertheless, this response was strongly coupled to the cloud IWP which is an indicator for the cloud thickness. In our case study, total surface precipitation rates are dominated by graupel which is not significantly affected by additional ship aerosol particles. Moreover, ship exhaust emissions have the potential to affect cloud radiative processes that play a vital role in the Arctic climate system. While the shortwave

radiative budget is mostly unaffected, ship exhaust perturbations can lead to both reductions and increases in net longwave radiative cooling at the surface and potentially impact the net surface radiative budget. The magnitude of the ship exhaust-induced cloud perturbations is strongly dependent on the number concentrations of the particle emissions. It is also affected more by the size of the exhaust particles than by their hygroscopicity. This means that the impact of ship emissions on Arctic cloud properties would depend strongly on the fuel types used and whether exhaust after-treatment systems like scrubbers are

used. Uncertainties in Arctic fuel type projections increase the challenge of constraining regional climate impacts from Arctic shipping. Transitions towards fuels with reduced sulfur content generally yield smaller CCN emissions, which potentially could reduce radiative effects from ship aerosol-cloud interactions. This is due to a reduction in hygroscopicity of the ship exhaust particles (Santos et al., 2023) and a shift in the particle size distribution towards smaller sizes (decreasing CMD; Lack et al., 2011; Yu et al., 2020, 2023). From June 2025 ships will no longer be allowed to carry and use fuel oils with densities and

viscosities exceeding predefined limits (IMO, 2021), which could have ramifications for wet scrubber usage, as these systems are mainly designed for use with high FSC residual fuels. Moreover, due to environmental concerns associated with increased Arctic shipping activity, Canada and several organizations have proposed black carbon emission control areas for Arctic waters, that include mandates for low FSC distillate fuel usage for ships operating in the Arctic (IMO, 2023b, a). Assuming that these proposals are ratified, they would limit the allowed Arctic ship fuels and thus, facilitate estimating the climate impact from

shipping. In this case, other environmental impacts from shipping such as BC deposition on snow surfaces, which reduces the surface albedo and enhances surface warming, could play a larger role and dominate the climate impact from Arctic shipping.

Ship exhaust particle concentrations above 1000 cm$^{-3}$ are realistic if one considers narrower and more localized regions where ship exhaust particles perturb clouds (Hobbs et al., 2000; Possner et al., 2018). Once emitted by a transiting ship, exhaust particles become dispersed in the atmosphere, resulting generally in smaller particle number concentrations. As a

result, perturbations would likely exert changes in cloud properties more akin to our low concentration model runs. One area of focus for future research could be to implement ship plume dispersion and use more realistic vertical exhaust particle concentration profiles. Moreover, exhaust particles will undergo chemical and physical transformations in the atmosphere associated with ship exhaust plume aging. In aged plumes, changes in particle size distributions are often observed due to coagulation of exhaust particles, and condensation and evaporation of water vapor and other atmospheric substances (Petzold

et al., 2008; Celik et al., 2020). If, for example, fewer but larger particles are present, it could have a stronger impact on cloud hydrometeors compared to the ship exhaust particle size distributions and number concentrations implemented in this study.



While this study focuses on Arctic shipping and clouds, IMO FSC regulations apply worldwide. This means that global ship exhaust emissions are subject to changes and thus, the global radiative forcing exerted by ship exhaust emissions will likely change. It is therefore important to improve our general understanding of the potential effects of FSC reduction and wet
scrubbing on particulate matter emissions and what this implies for cloud and climate processes.

## 5  Conclusions

In this study, we used LES together with aerosol data from laboratory experiments to examine the potential impact of ship exhaust particles on Arctic mixed-phase cloud properties. The laboratory experiments investigated the impacts of fuel sulfur content reduction and exhaust wet scrubbing on the physicochemical properties of ship exhaust particles (Santos et al.,
2022, 2023, 2024). Wet scrubbing and FSC reduction represent regulatory compliance measures in the maritime shipping sector, which affect ship exhaust particle emissions and could potentially be utilized by ships in the Arctic. Given the projected increase in Arctic shipping activity due to strongly declining Arctic sea-ice extent and the availability of shorter trans-Arctic transportation routes, we have sought to illuminate how ship emissions may impact Arctic clouds and thereby affect the regional radiative balance.

The simulations were done for a persistent stratiform mixed-phase cloud, based on observations from the ASCOS campaign (Tjernström et al., 2012, 2014) and previous simulations (Igel et al., 2017; Stevens et al., 2018; Christiansen et al., 2020; Sotiropoulou et al., 2021; Frostenberg et al., 2023). The simulated cloud was subsequently perturbed by adding ship exhaust particle profiles into the model domain. A selected number of model runs was repeated with increased pre-fixed $N_i$ to study the impact of ship exhaust perturbations on a thinner baseline cloud with increased IWP and reduced LWP.

Ship exhaust simulations revealed potential impacts on cloud droplet and raindrop concentrations, affecting the LWP and decreasing the cloud drop effective radius. Total surface precipitation was found to be mostly unaffected; liquid-phase precipitation was reduced, but it was only a minor constituent of total surface precipitation. Moreover, the cloud albedo increased marginally in all ship exhaust experiments. Our first set of simulations, with $N_i$ concentrations in line with observations and IWP values at the lower end of the retrieved values, demonstrated that ship exhaust perturbations can lead to a reduction in
longwave radiative cooling at the surface of up to 4.8 W m$^{-2}$. This result implies that ship emissions may lead to a net warming effect compared to our baseline simulation without ship exhaust aerosol. The magnitude of the surface radiation change depended on the hygroscopicity and the CMD of the added ship aerosol particles, where the effect of the CMD was most important. Additional sensitivity tests with $N_i$ increased to 600 m$^{-3}$, with reduced LWP and increased IWP (both in line with retrieved values), revealed that ship exhaust perturbations may lead to enhanced surface radiative cooling. This demonstrates
that the net effect of ship exhaust emissions on the radiative forcing exerted by Arctic low-level clouds would not only strongly depend on the prevalent fuel types, and whether ships in the Arctic utilize wet scrubbers for exhaust after-treatment, but also on the prevalent atmospheric conditions and cloud properties. Studies have shown that Arctic low-level cloud properties are strongly coupled to the surface properties, and that sea-ice free conditions can lead to generally larger cloud fractions and



increased LWP (Barton and Veron, 2012; Taylor and Monroe, 2023). It is therefore likely that future Arctic low-level cloud
properties may be more similar to our first case ($N_i = 200 \text{ m}^{-2}$).

Bulatovic et al. (2021) showed large variations in cloud microphysical properties for different background aerosol concentrations and sizes. Results of ship exhaust perturbations may therefore vary substantially with different background aerosol concentrations and thus, the microphysical structure of the perturbed cloud, as demonstrated by our additional set of simulations with increased $N_i$. The enhanced warming that the Arctic is experiencing will likely change the state of ambient aerosol
concentrations due to biogenic and anthropogenic processes (Schmale et al., 2021). For example, more open sea surface area will lead to enhanced new particle formation due to marine biogenic emissions (Dall´Osto et al., 2017). General low ambient aerosol number concentrations mean that already small increases in concentrations can have large impacts on cloud properties (Mauritsen et al., 2011). It is therefore expected that these changes will also affect the properties of Arctic clouds.

Our collective results do show evidence that Arctic shipping emissions can lead to alterations in the micro- and macrophys-
ical state of persistent Arctic low-level mixed-phase clouds. While a stronger tendency towards enhanced surface warming from ship exhaust emissions was obtained, this effect was mostly observed when ship aerosol concentrations were increased by $N = 1000 \text{ cm}^{-3}$. When low ship exhaust particle concentrations ($N = 100 \text{ cm}^{-3}$) were utilized, only wet scrubbing model runs were found to alter cloud radiative properties significantly compared to the baseline. However, given the ban on carriage and usage of high-density/viscosity residual fuel oils in 2025, wet scrubbing might not be utilized by a large fraction of ships
in the Arctic (IMO, 2021).

This study may help in constraining possible climate feedbacks from a projected increase in Arctic shipping activity. However, our results show that more information on future Arctic shipping activity, including fuel types, traffic volume and associated emissions characteristics, prevalent meteorological conditions, and cloud types is required for more accurate estimates.

*Data availability.* Model output data will be made available on an open access database. The experimental data utilized in this study has
been previously published and made available through the cited publications, Santos et al. (2022, 2023, 2024).




## Appendix A: Meteorological parameter of ASCOS case

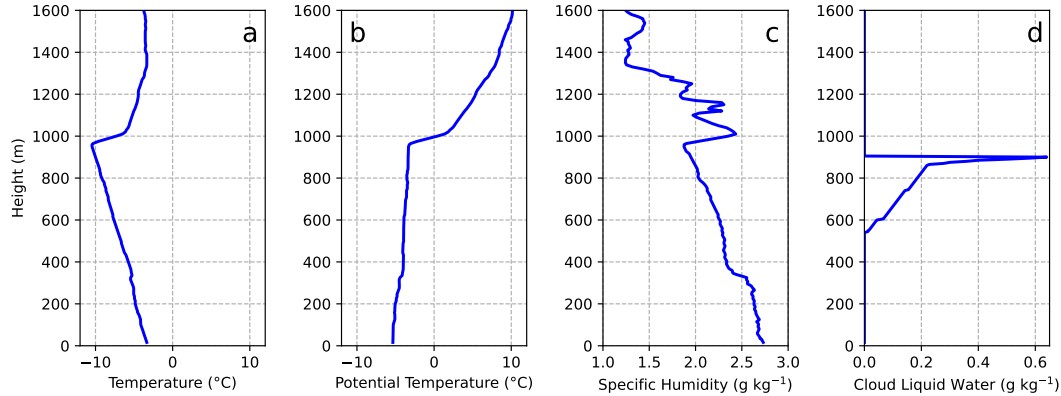

**Figure A1.** Radiosonde observations of (a) temperature, (b) potential temperature, (c) specific humidity and (d) derived cloud liquid water based on radiometer measurements performed on August 31st 2008 during the ASCOS campaign (Tjernström et al., 2012, 2014). The data was used to initialize MIMICA in this study.



## Appendix B: Profiles of ice-phase hydrometeors

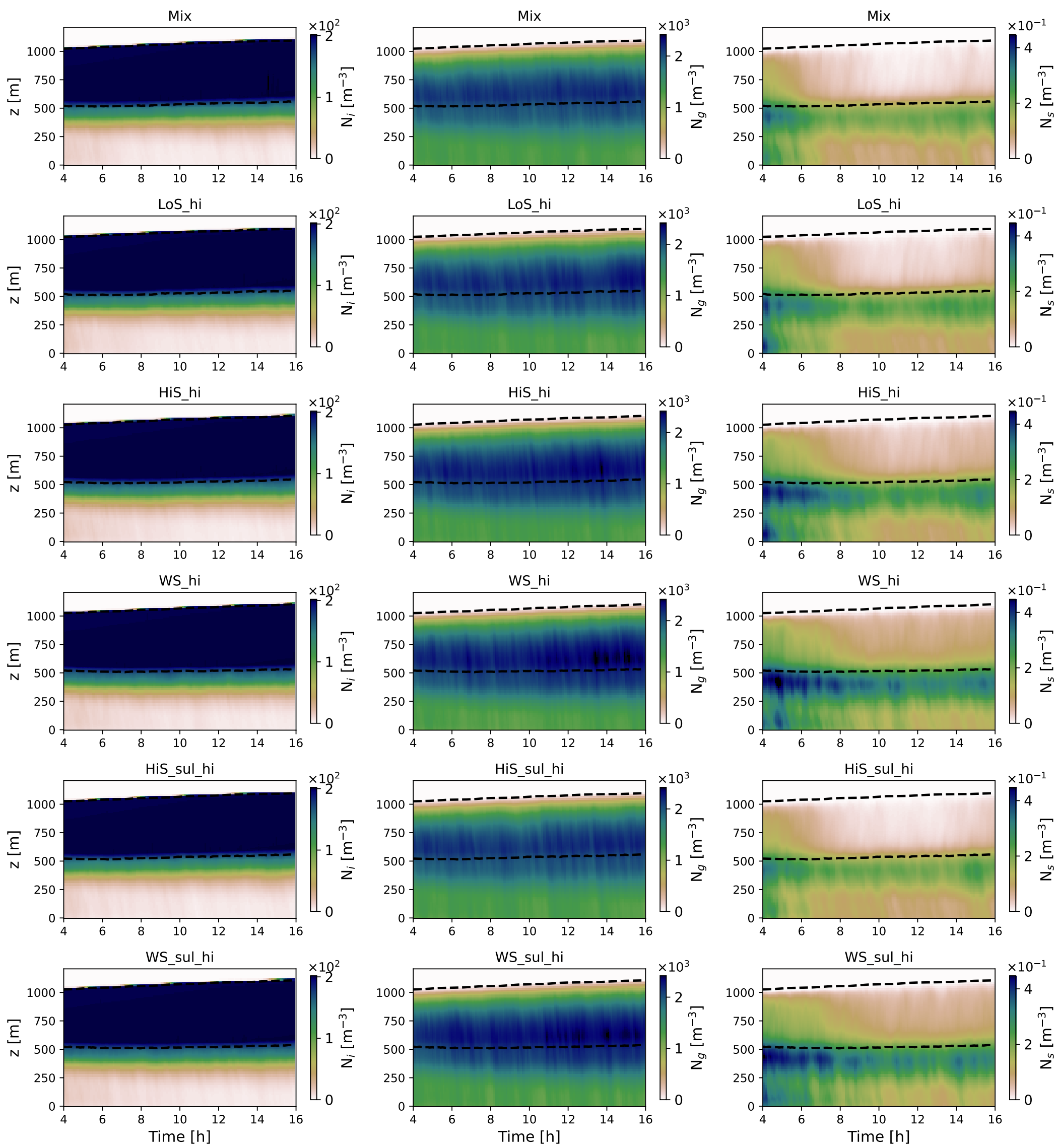

**Figure B1.** Temporal evolution of horizontally averaged number concentrations of ice crystals ($N_i$), graupel ($N_g$) and snow ($N_s$) simulated for the reference case (Mix) and the high ship aerosol concentration cases LoS_hi, HiS_hi, WS_hi, HiS_sul_hi and WS_sul_hi. The black dashed lines represent case-specific, horizontally averaged cloud bottom and cloud top heights. The spin-up period (0 to 4 h) is removed from all figures.



**Figure B2.** Temporal evolution of horizontally averaged mixing ratios of ice crystals ($Q_i$), graupel ($Q_g$) and snow ($Q_s$) simulated for the reference case (Mix) and the high ship aerosol concentration cases LoS_hi, HiS_hi, WS_hi, HiS_sul_hi and WS_sul_hi. The black dashed lines represent case-specific, horizontally averaged cloud bottom and cloud top heights. The spin-up period (0 to 4 h) is removed from all figures.



## Appendix C: Averaged profiles of ice-phase hydrometeors

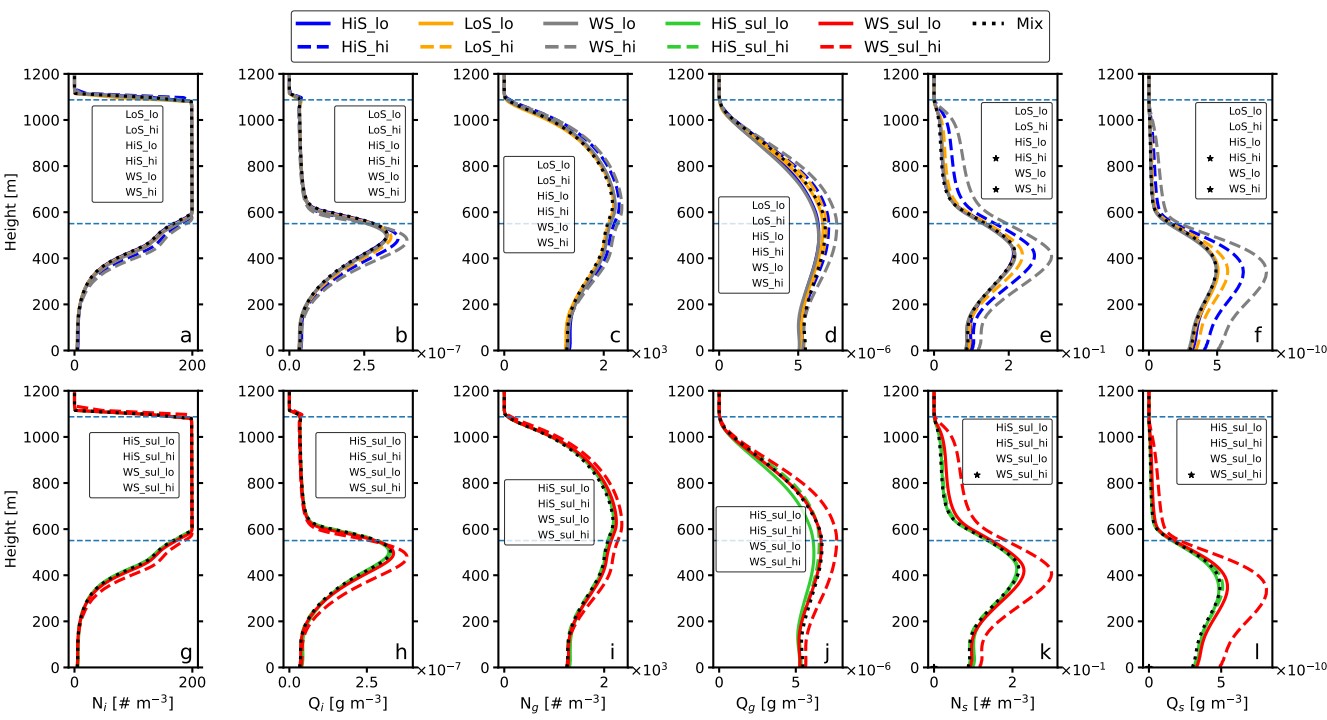

**Figure C1.** Vertical profiles of horizontally averaged (a and g) $N_i$, (b and h) $Q_i$, (c and i) $N_g$, (d and j) $Q_g$, (e and k) $N_s$ and (f and l) $Q_s$ averaged over the last four simulation hours. The light blue, dashed line represents the average cloud bottom and top height calculated for the reference case (Mix). HiS, LoS, and WS represent ship aerosol from measurements of high and low sulfur content fuels and wet scrubbing respectively (Santos et al., 2022, 2023). The HiS_sul and WS_sul cases represent sulfate particle modes of high FSC fuel combustion and exhaust gas wet scrubbing from Santos et al. (2024). The label additions _lo and _hi signify the ship aerosol concentrations used in the individual model runs. Significant differences between ship exhaust cases and Mix were assessed using two-sided $t$ tests at a confidence level of 95 %. Model runs with significant differences are marked with star icons in inset legends.

*Author contributions.* Idea: EST and AE. Conceptualization: LI, HF, AE. Model implementation and simulations: LS, HF, LI, and ABP.
Analysis of the data: LS, HF, LI, ABP. Visualization: LS. Writing, review and editing: All co-authors.

*Competing interests.* The authors declare that they have no conflict of interest.



*Acknowledgements.* This research was funded by the Swedish Research Councils FORMAS (2017-00564) and VR (2020-03497 & 2021-04042). EST, LS, HF, ABP, and LI thank the Swedish Strategic Research Area MERGE for their support. HF and LI were supported by Chalmers Gender Initiative for Excellence (Genie). The computations were enabled by resources provided by the National Academic Infrastructure for Supercomputing in Sweden (NAISS) at the National Supercomputer Centre (NSC) partially funded by the Swedish Research Council through grant agreement no. 2022-06725.




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
