# Peer review of "Potential impacts of marine fuel regulations on an Arctic stratocumulus case and its radiative response"

_EGUsphere, 2024_

## Referee Comment (RC1)

Reviewer comments dos Santos et al. 2024

The authors explore potential impacts of shipping activity on the properties and radiative effects of Arctic clouds. They use Large Eddy Simulations to investigate the effects of different fuel types and of emission management by scrubbing, as well as the effect of varying cloud conditions. This study is well set up and described, and I recommend it for publication after some minor comments are addressed.

General comments:

1. Non-cloud effects: Some broader overview (in the introduction) on the other potential impacts of arctic shipping on radiative forcing would be helpful. An order-of-magnitude estimate for the effect of, e.g., soot-on-snow albedo reduction, as well as direct radiative effects of aerosol in the arctic, or other effects, would be particularly valuable to situate the study in context and increase its value for non-cloud-scientists.

2. Semi-direct aerosol effect: Connected to the above point, in your simulations, radiation is not coupled to aerosol (l. 176). Can semi-direct effects of aerosol on clouds be excluded? What is the reason for not including this in the modelling?

3. Ice phase effects: You mention in the introduction, l. 95, that Christensen '14 and Possner '17 observe shifts to the ice phase from ship aerosols. In the methods, ll.168-171, you describe the choice of constant, diagnostic ice crystal number concentrations, motivated by the findings of ship aerosol as ineffective INPs. Is this not contradictory? In this setup, could it be misleading to write in the abstract (l. 15): "Simulated enhancements [...] predominantly affected the liquid phase properties of the cloud..." without referencing the diagnostic $N_i$ used (same in l. 372)?

Specific comments:

1. L. 240: Do you have a hypothesis for the mechanism for higher LWP with prescribed aerosol?

2. L. 322: "Similar relationships [...] *were* also noted by Christiansen et al. (2020).

3. L. 327: "Which is expected given the relatively large LWP" Is this because the albedo-LWP relationship saturates?

4. Fig.2: What is behind the periodic increases or bumps in IWP, which seem to come about earlier in the polluted cases than in "Mix"? Are they freezing events? The rain numbers seem to dip in the next figure, and in B2 the graupel has maxima... Do you think the warm phase changes could in turn make the polluted cases freeze out earlier?

5. Fig. 3: It takes my laptop a long time to render this figure, maybe you can rasterize it (also for the other heatmap figures)?

6. All figures: Green and red in the same panel (e.g. Fig.2) is not the most colourblind friendly choice. Consider changing.

---

## Author Response (AR1)

**Response to Reviewer Comments on *Potential impacts of marine fuel regulations on Arctic clouds and radiative feedbacks* by Santos et al. (2024)**

**September 10, 2024**

We thank both reviewers for their comments which have helped improve and clarify our manuscript. We have addressed the questions and comments made by the reviewers and submit a revised manuscript. Within this document reviewer comments are presented as *italicized* text, with our direct responses written in **bold** and associated changes to the manuscript highlighted in blue.

**Review 1**

*Reviewer comments dos Santos et al. 2024*

*The authors explore potential impacts of shipping activity on the properties and radiative effects of Arctic clouds. They use Large Eddy Simulations to investigate the effects of different fuel types and of emission management by scrubbing, as well as the effect of varying cloud conditions. This study is well set up and described, and I recommend it for publication after some minor comments are addressed.*

*General comments:*

*1. Non-cloud effects: Some broader overview (in the introduction) on the other potential impacts of arctic shipping on radiative forcing would be helpful. An order-of-magnitude estimate for the effect of, e.g., soot-on-snow albedo reduction, as well as direct radiative effects of aerosol in the arctic, or other effects, would be particularly valuable to situate the study in context and increase its value for non-cloud-scientists.*

**We have added a paragraph in the introduction which addresses the reviewer's comment:**

Ship exhaust emissions also have the potential to exert radiative forcing via direct interactions between emitted particles and solar radiation, or via reduction of the surface albedo due to deposition of light-absorbing black carbon (BC) particles onto snow. A modelling study by Dalsøren et al. (2013) examined a number of direct and indirect processes related to shipping emissions and radiative processes. The study found significant seasonal variability for all processes and that direct sulfate aerosol interactions exert the largest radiative forcing (positive) out of all processes, i.e., a larger forcing than aerosol-cloud interactions. Given IMO's marine fuel policies, the impact of ship-related sulfate contributions may be subject to large uncertainties. In contrast, Gilgen et al. (2018) and Stephenson et al. (2018) found that radiative forcing induced by aerosol-cloud interactions outweighs forcing exerted by direct aerosol-radiation interaction and BC deposition onto snow. Similarly, Browse et al. (2013) and Li et al. (2021) report only minor contributions of BC deposition from shipping activity which would yield insignificant changes in radiative forcing and not contribute to accelerated sea ice loss. While these finding apply for the Arctic in general, surface albedo adjustments due to BC deposition may have stronger local constrained impacts, for example, in the sub-Arctic region (Browse et al., 2013).

*2. Semi-direct aerosol effect: Connected to the above point, in your simulations, radiation is not coupled to aerosol (l. 176). Can semi-direct effects of aerosol on clouds be excluded? What is the reason for not including this in the modelling?*

**Semi-direct aerosol effects cannot be excluded, and are very complex to simulate. The model version used in this study does not have aerosol-radiation interactions implemented. An implementation of respective processes is currently in development.**

*3. Ice phase effects: You mention in the introduction, l.95, that Christensen'14 and Possner'17 observe shifts to the ice phase from ship aerosols. In the methods, ll.168-171, you describe the choice of constant, diagnostic ice crystal number concentrations,*

*motivated by the findings of ship aerosol as ineffective INPs. Is this not contradictory? In this setup, could it be misleading to write in the abstract (l. 15): "Simulated enhancements [...] predominantly affected the liquid phase properties of the cloud..." without referencing the diagnostic Ni used (same in l. 372)?*

MIMICA includes a large number of options for the implementation of aerosol particles and microphysical processes. The decision to utilize diagnostic ice crystal number concentrations was both motivated by our findings regarding the ineffectiveness of ship exhaust particle to act as INP (as correctly pointed out by the reviewer) and due to it being a well established method already used with MIMICA and the ASCOS case (e.g., Stevens et al. (2018); Bulatovic et al. (2021); Frostenberg et al. (2023)). Regarding Christensen et al. (2014) it is important to highlight that the authors observed signals of increased ice fractions in ship-exhaust polluted clouds for only one of the two employed methods. Furthermore, the authors point out that a simultaneous increase in cloud droplet numbers may have increased the noise in the retrievals potentially increasing uncertainties (Christensen et al., 2014). In both cases meteorological (e.g., inversion layer height, and temperature and humidity profiles) and macrophysical cloud properties (e.g., cloud thickness and hydrometeor background concentrations) varied substantially compared to the ASCOS case (Christensen et al., 2014; Possner et al., 2017). Possner et al. (2017), for example, utilized INP background concentrations of 2 - 5 $L^{-1}$ and varied additional shipping-related INP between 0 and 5 $L^{-1}$ (for comparison, our study, as well as the aforementioned MIMICA-related studies, utilized ice crystal number concentrations of 0.2 $L^{-1}$). All these differences in observational data and implementation of numerical methods add to uncertainty of the outcome and should be considered. L. 15 in the abstract has been rephrased to:

Simulations with diagnostic ice crystal number concentrations revealed that enhancements of ship exhaust particles predominantly affected the liquid-phase properties of the cloud [...] .

*Specific comments:*

*1. L.240: Do you have a hypothesis for the mechanism for higher LWP with prescribed aerosol?*

This is due to the absence of aerosol removal upon activation into droplets when aerosol are fully prescribed as in this study. In Stevens et al. (2018), the authors report that precipitation formation and thus, removal of aerosol to the surface, are depressed with increasing initial aerosol concentrations. This drizzle suppression allows for a build up of liquid cloud water.

*2. L.322: "Similar relationships [...] were also noted by Christiansen et al. (2020).*

Corrected. The sentence now reads:

Similar relationships between increased LWP and a reduced LW radiative cooling were also noted by Christiansen et al. (2020).

*3. L.327: "Which is expected given the relatively large LWP" Is this because the albedo-LWP relationship saturates?*

**We have expanded our discussion regarding changes in SW surface fluxes. The corresponding paragraph in Section 3.3 now reads:**

The net SW radiation is positive in all simulations, meaning the net flux is downwelling. In all simulations, the net SW fluxes initially increase until 6 h into the respective simulations where a maximum of around 14 W m$^{-2}$ is reached. By the end of the simulations, net SW decreases to $\approx$5 W m$^{-2}$. The temporal trends in LW and SW radiation both coincide with the solar angle. The results indicate that WS cases tend to slightly decrease the net SW (Table 2 and Figure C1), yet, none of the ship sensitivity tests are found to significantly impact net SW fluxes at the surface, despite associated increases in $\alpha$ (Fig.5 i - j). Changes in cloud properties induced by ship exhaust perturbations are expected to only lead to small changes in SW surface fluxes, due to the reduced solar fluxes based on the geographical location, and the comparatively large LWP, which leads to a substantial extinction of incoming SW radiation. Relatively small changes in $\alpha$ are therefore only expected to lead to minor changes in SW surface fluxes.

**Moreover, we add two additional figures to the appendix showing relative changes in net SW surface fluxes between high ship exhaust particle concentration cases and the respective baseline/reference cases (see Figures C1 and C2).**

**A rounding mistake was found in the calculation of mean net SW values. The net SW value of WS_hi has been changed from 6.4 to 6.3 in Table 2.**

*4. Fig.2: What is behind the periodic increases or bumps in IWP, which seem to come about earlier in the polluted cases than in "Mix"? Are they freezing events? The rain numbers seem to dip in the next figure, and in B2 the graupel has maxima... Do you think the warm phase changes could in turn make the polluted cases freeze out earlier?*

**Indeed, these sporadic spikes in IWP are associated with increased formation of graupel. The same phenomena has also been observed by Bulatovic et al. (2021), who used a very similar setup. Therein, the authors state that this is a result of strong collection rates of raindrops by graupel. In section 3.1 we added:**

Sporadic spikes in the temporal evolution of IWP are in all cases caused by increased graupel formation rates at the expense of raindrops. Similar features in IWP evolution are reported by Bulatovic et al. (2021) who used MIMICA with a similar setup.

*5. Fig.3: It takes my laptop a long time to render this figure, maybe you can rasterize it (also for the other heatmap figures)?*

**Fig. 3, B1 and B2 have been converted to PNG files reducing the manuscript file size and figure rendering times.**

*6. All figures: Green and red in the same panel (e.g. Fig.2) is not the most colourblind friendly choice. Consider changing.*

**Thank you for bringing this to our attention. We changed the color of all HiS_sul cases to a more yellowish hue and updated all figures.**

**Review 2**

*The manuscript explores the impact of ship emissions to a campaign-based Arctic cloud scene through large-eddy simulations. To mimic various ship exhaust technologies and their emissions, the authors impose aerosol particle size distributions informed by previous laboratory work and find microphysical responses that slightly alter clouds' condensate amount and emitted longwave radiation.*

*The overall manuscript is well written. I have a couple of concerns that the authors should address before publication.*

*Major concerns*

*The title suggests a wider picture on Arctic clouds, but the manuscript focuses on a particular case. It is unclear, whether these results are representative for the wider Arctic. The authors should (1) put the selected case into perspective by using cloud statistics from Arctic observations, and (2) discuss potential impacts to clouds that have not been touched on here. I wonder if there are other cases of stronger cloud-aerosol-precipitation interaction, where the various aerosol scenarios make a more meaningful difference.*

**The reviewer raises a valid concern. Whereas the ASCOS case used in this study is based on observations made in the high Arctic, increased Arctic shipping activity will mostly occur in coastal regions, e.g., the Northeast Passage and the Northern Sea Route. This implies that ship exhaust emissions would perturb clouds of potentially different characteristics. This is one of the study's limitations because the model has not been adjusted or tested for such a case, yet. More observational data is required, which was beyond the scope of this study. We address this point and add to the conclusions:**

Consequently, it is important to highlight that the case study used in this study is based on observations made in the high Arctic. Most of Arctic shipping activity will likely occur closer to coastal regions where air masses are likely to be more strongly influenced by anthropogenic and biogenic activity (see, for example, Smith and Stephenson (2013)). This means that the atmospheric background conditions and cloud properties may vary from the mixed-phase cloud case studied here and will likely affect the impact of ship exhaust perturbations on cloud properties.

*Perhaps it also necessary to change the title to: "Potential impacts of marine fuel regulations on an Arctic stratocumulus case and its radiative response".*

**We agree that the manuscript title should more clearly reflect that this manuscript focuses on a specific Arctic cloud case which cannot not be used to generalize to all Arctic cloud types and therefore, cloud responses from ship exhaust perturbations. We think that the title suggested by the reviewer is an adequate adjustment and decided to change the title of the manuscript to:**

Potential impacts of marine fuel regulations on an Arctic stratocumulus case and its radiative response

*The description of simulations lacks important details: (1) Looking at Christiansen et al., (2020), there is a large-scale divergence imposed – is that the case here, too?*

The same large-scale divergence rate of $1.5 \times 10^{-6}$ s$^{-1}$ is used in this study. We have expanded Section 2.2 by adding more information regarding the simulation setup:

The radiation solver used in this study is based on Fu and Liou (1992). It is important to note that while radiation is affected by cloud hydrometeors it is not affected by aerosols. Surface temperature and pressure have prescribed values of 269.8 K and 1026.3 hPa, respectively. The surface albedo is set to 0.844 and the surface roughness to 0.0004 m. Sensible and latent heat fluxes at the surface are both set to 0 W m$^{-2}$ based on the small values reported in Tjernström et al. (2014). A large-scale divergence of $1.5 \times 10^{-6}$ s$^{-1}$ is imposed over the whole domain. Large-scale advection is turned off in the model.

*(2) Furthermore, the authors show the vertical temperature profile and indicate that it's kept constant – is that achieved through nudging or advective tendencies and is either technique applied throughout or only in the free troposphere?*

Thank you for the comment, this was an unfortunate phrasing. Temperature actually does not stay constant throughout the simulation, but is rather a prognostic variable that is influenced by sources calculated in the model (radiation, latent heating/cooling, turbulent diffusion) and the large-scale divergence mentioned above. This is the case within the model domain. Above the domain, a constant temperature profile is used only for the radiation solver and this is literally constant throughout the simulation. We have clarified this in the manuscript:

Note that meteorological conditions from ASCOS were only used to initialize the model. Potential temperature and total water mixing ratio are prognostic variables influenced by sources and sinks in the model (e.g., radiation, microphysical phase changes, and precipitation), but do not necessarily represent the temporal evolution of the real atmospheric state.

*(3) The authors should also specify how they calculate turbulent surface fluxes.*

See response to the question regarding the large-scale divergence above.

*(4) How do the authors justify imposing ship-based aerosol uniformly rather than only within the marine boundary layer?*

It is true that assuming uniform ship-based aerosol may not be the most realistic representation of how ship exhaust emissions may perturb the stratocumulus cloud in this study. Initial tests with vertically constrained ship exhaust aerosol profiles revealed that the updraft in the domain was insufficient to fully mix aerosol particles into the cloud layer and therefore, inadequate to investigate the premise of the study. It should be noted that similar approaches were utilized in similar studies (e.g., Possner et al. (2017) and Eirund et al. (2019)). Ideally, ship exhaust plume height and dispersion estimations would yield the most realistic representation of ship exhaust perturbations. Nevertheless, this was not in the scope of this study but should be considered for future studies. This was addressed in the original manuscript (see ll. 393 - 397).

*The authors show a significant difference in longwave emissions but none in the shortwave spectrum. The latter is puzzling to me, and the authors should elaborate on why that is – are there perhaps compensating effects of cloud fraction and cloud albedo?*

**This question was also lifted by Reviewer 1. Please see our response to question 3 of Review 1, under "Specific comments".**

*Minor concerns*

*ll. 108-110 Would it be possible to provide satellite imagery of this case (or at least coordinates) so that readers can obtain a visual impression of the case?*

**In ll. 177 - 178 we add:**

The simulated stratocumulus case is based on observations made during ASCOS on 31.08.2008 at approximately 87°N, 11°W [...]

*ll. 136ff (and also Fig. 1) It took me a while to understand the scenarios and their number of modes (and would expect the same for other readers). Perhaps it would be simpler to display all initial aerosol size distributions in Fig 1. I'm also not sure I understand the value of Fig. 1b – perhaps its better suited in the appendix?*

**In order to emphasize on the fact that engine experiments were performed using different engines and fuel types we added in ll. 142-143:**

Engine experiments summarized in Santos et al. (2024) utilized a different engine with higher power output and fuels with different properties compared to Santos et al. (2022, 2023) and therefore, resulted in different emission characteristics.

**We add the corresponding references to Fig. 1 a and b to make it more clear. Moreover, the colors of the individual particle size distributions and modes in Fig. 1 have been adjusted to match the case-specific color coding used in later figures.**

*l. 191 The label "mix" is confusing and should instead be labelled as "no ship".*

**We agree with this statement. The "Mix" label has been renamed to "no_ship". The text and all figures have been adjusted correspondingly.**

*l. 381 "smaller" is ambiguous here.*

**The sentence has been rephrased and now reads:**

Transitions towards fuels with reduced sulfur content have been shown to lead to substantial reductions in CCN number emissions, which potentially could reduce radiative effects from ship aerosol-cloud interactions.

**References**

J. Browse, K. S. Carslaw, A. Schmidt, and J. J. Corbett. Impact of future Arctic shipping on high-latitude black carbon deposition. *Geophysical Research Letters*, 40(16):4459–4463, 2013. doi: 10.1002/grl.50876.

I. Bulatovic, A. L. Igel, C. Leck, J. Heintzenberg, I. Riipinen, and A. M. L. Ekman. The importance of Aitken mode aerosol particles for cloud sustenance in the summertime high Arctic – a simulation study supported by observational data. *Atmos. Chem. Phys.*, 21(5):3871–3897, 2021. doi: 10.5194/acp-21-3871-2021.

M. W. Christensen, K. Suzuki, B. Zambri, and G. L. Stephens. Ship track observations of a reduced shortwave aerosol indirect effect in mixed-phase clouds. *Geophysical Research Letters*, 41(19):6970–6977, 2014. ISSN 19448007. doi: 10.1002/2014GL061320.

S. Christiansen, L. Ickes, I. Bulatovic, C. Leck, B. J. Murray, A. K. Bertram, R. Wagner, E. Gorokhova, M. E. Salter, A. M. Ekman, and M. Bilde. Influence of Arctic Microlayers and Algal Cultures on Sea Spray Hygroscopicity and the Possible Implications for Mixed-Phase Clouds. *Journal of Geophysical Research: Atmospheres*, 125(19), 2020. doi: 10.1029/2020JD032808.

S. B. Dalsøren, B. H. Samset, G. Myhre, J. J. Corbett, R. Minjares, D. Lack, and J. S. Fuglestvedt. Environmental impacts of shipping in 2030 with a particular focus on the Arctic region. *Atmospheric Chem. Phys.*, 13(4):1941–1955, 2013. ISSN 1680-7316. doi: 10.5194/acp-13-1941-2013.

G. K. Eirund, A. Possner, and U. Lohmann. Response of Arctic mixed-phase clouds to aerosol perturbations under different surface forcings. *Atmospheric Chemistry and Physics*, 19(15):9847–9864, 2019. ISSN 1680-7324. doi: 10.5194/acp-19-9847-2019.

H. C. Frostenberg, A. Welti, M. Luhr, J. Savre, E. S. Thomson, and L. Ickes. The chance of freezing – a conceptional study to parameterize temperature-dependent freezing by including randomness of ice-nucleating particle concentrations. *Atmospheric Chemistry and Physics*, 23(19):10883–10900, 2023. doi: 10.5194/acp-23-10883-2023.

Q. Fu and K. N. Liou. On the correlated k-distribution method for radiative transfer in nonhomogeneous atmospheres. *Journal of Atmospheric Sciences*, 49(22):2139 – 2156, 1992. doi: 10.1175/1520-0469(1992)049⟨2139:OTCDMF⟩2.0.CO;2.

A. Gilgen, W. T. K. Huang, L. Ickes, D. Neubauer, and U. Lohmann. How important are future marine and shipping aerosol emissions in a warming Arctic summer and autumn? *Atmospheric Chemistry and Physics*, 18(14):10521–10555, 2018. ISSN 1680-7324. doi: 10.5194/acp-18-10521-2018.

X. Li, A. H. Lynch, D. A. Bailey, S. R. Stephenson, and S. Veland. The impact of black carbon emissions from projected Arctic shipping on regional ice transport. *Climate Dynamics*, 57(9-10):2453–2466, 2021. doi: 10.1007/s00382-021-05814-9.

A. Possner, A. M. L. Ekman, and U. Lohmann. Cloud response and feedback processes in stratiform mixed-phase clouds perturbed by ship exhaust. *Geophysical Research Letters*, 44(4):1964–1972, 2017. ISSN 00948276. doi: 10.1002/2016GL071358.

L. C. Smith and S. R. Stephenson. New Trans-Arctic shipping routes navigable by midcentury. *Proceedings of the National Academy of Sciences*, 110(13), 2013. doi: 10.1073/pnas.1214212110.

S. R. Stephenson, W. Wang, C. S. Zender, H. Wang, S. J. Davis, and P. J. Rasch. Climatic Responses to Future Trans-Arctic Shipping. *Geophysical Research Letters*, 45 (18):9898–9908, 2018. doi: 10.1029/2018GL078969.

R. G. Stevens, K. Loewe, C. Dearden, A. Dimitrelos, A. Possner, G. K. Eirund, T. Raatikainen, A. A. Hill, B. J. Shipway, J. Wilkinson, S. Romakkaniemi, J. Tonttila, A. Laaksonen, H. Korhonen, P. Connolly, U. Lohmann, C. Hoose, A. M. L. Ekman, K. S. Carslaw, and P. R. Field. A model intercomparison of CCN-limited tenuous clouds in the high Arctic. *Atmospheric Chemistry and Physics*, 18(15):11041–11071, 2018. doi: 10.5194/acp-18-11041-2018.

M. Tjernström, C. Leck, C. E. Birch, J. W. Bottenheim, B. J. Brooks, I. M. Brooks, L. Bäcklin, R. Y.-W. Chang, G. de Leeuw, L. Di Liberto, S. de la Rosa, E. Granath, M. Graus, A. Hansel, J. Heintzenberg, A. Held, A. Hind, P. Johnston, J. Knulst, M. Martin, P. A. Matrai, T. Mauritsen, M. Müller, S. J. Norris, M. V. Orellana, D. A. Orsini, J. Paatero, P. O. G. Persson, Q. Gao, C. Rauschenberg, Z. Ristovski, J. Sedlar, M. D. Shupe, B. Sierau, A. Sirevaag, S. Sjogren, O. Stetzer, E. Swietlicki, M. Szczodrak, P. Vaattovaara, N. Wahlberg, M. Westberg, and C. R. Wheeler. The Arctic Summer Cloud Ocean Study (ASCOS): Overview and experimental design. *Atmospheric Chem. Phys.*, 14(6):2823–2869, 2014. ISSN 1680-7324. doi: 10.5194/acp-14-2823-2014.

---

## Author Response (AR2)

**Response to Reviewer Comments on *Potential impacts of marine fuel regulations on an Arctic stratocumulus case and its radiative response* by Santos et al. (2024)**

October 3, 2024

We thank both reviewers for their comments which have helped improve and clarify our manuscript. Within this document reviewer comments are presented as *italicized* text, with our direct responses written in **bold** and associated changes to the manuscript highlighted in blue.

**Review 1**

*The authors have addressed all of my comments and I recommend publication.*

*Technical corrections: - l. 116 "findings"*

**Thank you for bringing this to our attention. The typo is now corrected.**

**General Comments**

**In the Data Availability section we changed "The experimental data utilized in this study has been previously published [...]." to**

The experimental data utilized in this study have been previously published [...].